# Understanding Bias Reinforcement in LLM Agents Debate

Jihwan Oh [* 1]  Minchan Jeong [* 1]  Jongwoo Ko [† 1]  Se-Young Yun [† 1]

## Abstract

Large Language Models (LLMs) solve complex problems using training-free methods like prompt engineering and in-context learning, yet ensuring reasoning correctness remains challenging. While self-correction methods such as self-consistency and self-refinement aim to improve reliability, they often reinforce biases due to the lack of effective feedback mechanisms. Multi-Agent Debate (MAD) has emerged as an alternative, but we identify two key limitations: bias reinforcement, where debate amplifies model biases instead of correcting them, and lack of perspective diversity, as all agents share the same model and reasoning patterns, limiting true debate effectiveness. To systematically evaluate these issues, we introduce *MetaNIM Arena*, a benchmark designed to assess LLMs in adversarial strategic decision-making, where dynamic interactions influence optimal decisions. To overcome MAD's limitations, we propose **DReaMAD** (**D**iverse **Rea**soning via **M**ulti-**A**gent **D**ebate with Refined Prompt), a novel framework that (1) refines LLMs' strategic prior knowledge to improve reasoning quality and (2) promotes diverse viewpoints within a single model by systematically modifying prompts, reducing bias. Empirical results show that **DReaMAD** significantly improves decision accuracy, reasoning diversity, and bias mitigation across multiple strategic tasks, establishing it as a more effective approach for LLM-based decision-making.

## 1. Introduction

Large Language Models (LLMs) have demonstrated remarkable problem-solving capabilities across a wide range of tasks by leveraging knowledge acquired from vast datasets (Achiam et al., 2023; The; Team et al., 2023; Dubey et al., 2024). These models can address complex decision-making problems using training-free methods such as *prompt engineering* (White et al., 2023; Chen et al., 2023a; Schulhoff et al., 2024b) and *in-context learning* (Brown et al., 2020; Dong et al., 2022; Wei et al., 2022; Mavromatis et al., 2023; Pan, 2023), which provide guidance for effective reasoning. However, these approaches do not explicitly guarantee the correctness of the generated responses.

To address this, recent research has explored self-correction mechanisms that allow LLMs to refine their own outputs without external feedback. Self-consistency (Wang et al., 2022; Chen et al., 2023c) enhances reliability by ensembling multiple responses. Self-refinement (Wan et al., 2023; Shinn et al., 2023; Madaan et al., 2024) enables LLMs to iteratively critique and revise their outputs. However, self-consistency lacks a critical feedback mechanism, meaning it does not iteratively refine responses but merely reduces the model's inherent randomness by converging on the most frequently generated answer. Further recent studies (Huang et al., 2024) suggest that self-refinement can degrade performance, as models often struggle to assess the correctness of their own reasoning.

Recently, inspired by the *Society of Mind* philosophy, Multi-Agent Debate (MAD; Chan et al. 2023; Du et al. 2023; Liang et al. 2023) has emerged as a promising alternative. However, its success has been limited to static problem-solving and lacks assessments for adversarial strategic reasoning (Cobbe et al., 2021; Edwards, 1994; He et al., 2020). Additionally, current evaluations do not account for dynamic decision-making in interactive environments, where an agent's choices influence and adapt to an opponent's actions. This limitation hinders LLMs from retrieving and applying strategic knowledge beyond the given context.

To overcome the above limitation, we introduce *MetaNIM Arena*, a framework for evaluating LLMs in adversarial strategic decision-making. It allows us to assess their ability to adapt dynamically and ensures robust reasoning under mathematically rigorous conditions. Through *MetaNIM Arena*, we systematically analyze two fundamental limitations of MAD:

**(1) Bias Reinforcement:** In strategic reasoning tasks, LLMs tend to rely on immediate context rather than

---
*Equal contribution  [1]KAIST AI, Seoul, Republic of Korea.
†Correspondence to: Jongwoo Ko <jongwoo.ko@kaist.ac.kr>,
Se-Young Yun <yunseyoung@kaist.ac.kr>.

*Proceedings of the 42nd International Conference on Machine Learning*, Vancouver, Canada. PMLR 267, 2025. Copyright 2025 by the author(s).

*Table 1.* Feature comparison between self-correction methods. In contrast to Multi-Agent Debate, `DReaMAD` encourages diverse viewpoints by varying prompts and enhancing debate robustness through automated knowledge structuring.

| Methods | Single Model Usage | Multiple Instances | Rethinking Process | Self-Feedback | Diversity in reasoning | Perspective Shifts |
|---|---|---|---|---|---|---|
| Self-Consistency | ✓ | ✓ | ✗ | ✗ | △ | ✗ |
| Self-Refinement | ✓ | ✗ | ✓ | ✓ | ✗ | ✗ |
| Multi-Agent Debate | ✓ | ✓ | ✓ | ✗ | △ | ✗ |
| **DReaMAD** (Ours) | ✓ | ✓ | ✓ | ✗ | ✓ | ✓ |

△: means diversity from LLM's randomness, controlled by *temperature* hyperparameter.

retrieving broader strategic knowledge, leading to distorted reasoning instead of correct inference. Debate-based frameworks further amplify this issue by reinforcing the model's inherent biases rather than mitigating them (§4.1-4.2).

**(2) Lack of Perspective Diversity:** Although MAD uses a debate structure, it relies on multiple instances of the same model. This limits the diversity of perspectives introduced in the reasoning process, reducing its ability to challenge inherent biases (§4.3).

Rooted in the *Learning from Multiple Approaches* framework (Council et al., 2005; Cleaves, 2008), research shows that engaging with multiple problem-solving representations enhances comprehension and mitigates biases. Building on this insight, we propose **DReaMAD** (**D**iverse **Rea**soning via **M**ulti-**A**gent **D**ebate with Refined Prompt). **DReaMAD** addresses the limitations of MAD by (1) refining LLMs' domain-specific knowledge to guide more accurate strategic reasoning, and (2) systematically modifying prompts to foster diverse perspectives. A detailed comparison with existing self-correction methods is presented in Table 1. Our key contributions are as follows:

- We introduce **MetaNIM Arena**, a benchmark designed to evaluate LLMs in adversarial strategic decision-making, where mathematical rigor enables precise assessment of reasoning quality and strategic adaptability.

- We identify the **bias reinforcement** in Multi-Agent Debate, showing that MAD strengthens both correct and incorrect reasoning rather than inherently improving it.

- We propose **DReaMAD**, a novel framework that refines strategic prior knowledge and enhances reasoning diversity through structured self-prompt refinement and perspective diversification, achieving a +12.0% accuracy gain over standard prompting on *MetaNIM Arena* dataset and a +20.8% higher win rate than MAD in the simulator.

## 2. Preliminary

### 2.1. Bias in LLMs

Large Language Models (LLMs) can exhibit biases that lead to unfair or skewed outcomes, arising from training data, model architectures, learning objectives, or deployment conditions (Guo et al., 2024). Such biases manifest both *intrinsically*, for instance in word embeddings (Bolukbasi

et al., 2016), and *extrinsically*, reflecting real-world disparities (Goldfarb-Tarrant et al., 2021). Moreover, biases can emerge dynamically during interactive reasoning, where current reinforcement mechanisms—like self-consistency and self-refinement—often fail to mitigate them (Huang et al., 2024; Shin et al., 2024). Indeed, recent research shows that iterative interactions can reinforce existing biases instead of diversifying reasoning (Ganguli et al., 2023).

### 2.2. Prompt Engineering and Self-Correction in LLMs

Prompt engineering shapes model outputs without retraining, potentially improving generalization and reducing bias (Brown et al., 2020; Reynolds & McDonell, 2021; Zhao et al., 2024; Schulhoff et al., 2024a; Shin et al., 2024). However, fully eliminating biases in complex reasoning remains challenging (Jiang et al., 2022; Lu et al., 2022).

Meanwhile, self-correction mechanisms in LLMs refine responses without external supervision (Ganguli et al., 2023; Liu et al., 2024; Kamoi et al., 2024). Self-consistency, for instance, ensembles multiple outputs but converges on frequent rather than correct answers (Wang et al., 2022; Chen et al., 2023c), and self-refinement can reinforce rather than fix biases (Wan et al., 2023; Shinn et al., 2023; Madaan et al., 2024; Huang et al., 2024). Feedback-loop methods such as STaR (Zelikman et al., 2022), Reflexion (Shinn et al., 2023), and SCoRe (Kumar et al., 2024) also struggle to reliably correct biases or foster diverse reasoning (Guo et al., 2024).

### 2.3. Multi-Agent Debate in LLMs

Multi-Agent Debate (MAD) enables LLM agents to critique each other, enhancing reasoning on complex tasks (Liang et al., 2023; Du et al., 2023). ChatEval (Chan et al., 2023), a multi-agent evaluation system, simulates human judgment to assess model output quality. Optimizations include task-specific strategies for improving debate effectiveness (Smit et al., 2024) and ACC-Debate, an actor-critic framework that trains models to specialize in debates, achieving benchmark gains (Estornell et al., 2024). While these enhancements improve performance, studies reveal a key limitation: static evaluations focus on assessing predefined problems, whereas real-world decision-making often involves dynamic, interactive environments where biases can evolve. Understanding how biases shift in these settings is crucial for developing robust strategies that extend beyond conventional static benchmarks.

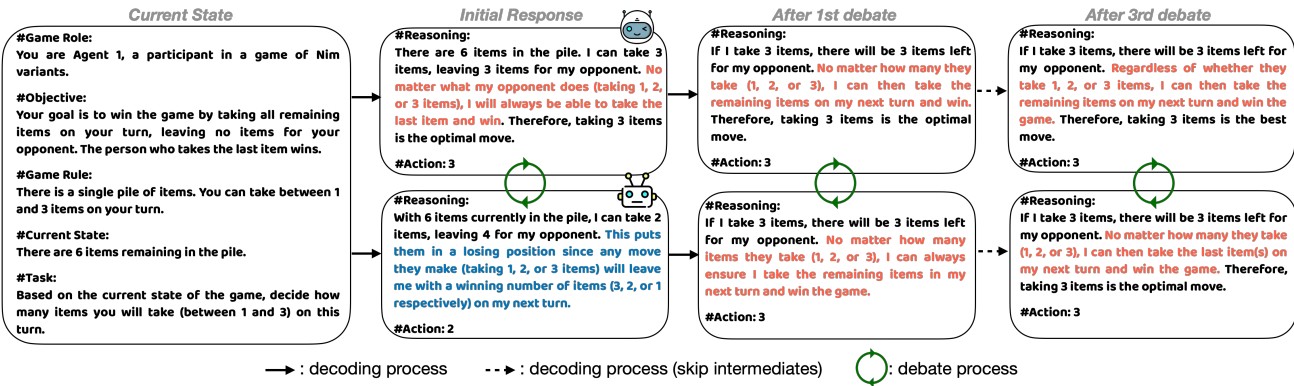

*Figure 1.* An example demonstrating how the debate process converges to a biased outcome. We observed that bias reinforcement occurs in the first debate. Blue text indicates the correct reasoning and orange text indicates the strong consistent (biased) reasoning. The second debate is omitted, as its procedure replicates the first and third; all debates use `GPT-4o-mini` as the debating agent.

## 3. MetaNIM Arena

### 3.1. Overview

We introduce *MetaNIM Arena*, illustrated in Figure 7. It features six impartial combinatorial games, meaning both players share identical moves at each state, all information is fully observable, and each game terminates in a finite number of moves. By merging *combinatorial game theory* with adversarial play, *MetaNIM Arena* serves as a benchmark for debate-based strategic reasoning in LLMs.

**Dataset and Simulator.** Key game situations are systematically collected into a *dataset*, each accompanied by an optimal action, enabling LLMs to be tested for decision-making accuracy. These results appear in Table 3, with details in Appendix A.3. Separately, *MetaNIM Arena* can function as a *simulator*: the model encounters an adaptive opponent, so each trajectory depends on both the agent's and the opponent's actions. Here, binary win/loss outcomes allows evaluation by *win rate*. We demonstrate this approach in Table 4, with further explanation in Appendix A.4. We refer the Appendix B for details of opponent modeling.

### 3.2. Why MetaNIM Arena?

*MetaNIM Arena* provides a rigorous environment for *adversarial strategic reasoning* in LLMs. Rather than isolated problem-solving or factual recall, our framework uses impartial combinatorial games, where each position's Grundy number defines the provably correct move. We will discuss further theoretical details in the following Section 3.4. This design offers:

**1. Adversarial Strategic Reasoning:** Each scenario includes an opponent whose actions shape outcomes. Models must *anticipate* adversarial moves across multiple turns, going beyond static QA or single-step predictions. This approach tests *latent strategic knowledge* in an interactive, step-by-step context.

**2. Clear Optimality Criterion:** By the Sprague-Grundy

Theorem, these games admit an optimal strategy. *MetaNIM Arena* thus measures how closely a model's reasoning aligns with that strategy, instead of relying on approximate metrics like BLEU or perplexity. We also assert that *MetaNIM Arena* naturally supports reinforcement learning framework. By providing a binary win/loss signal and structuring gameplay as a Markov Decision Process (MDP), it enables iterative strategy refinement—an advantage often absent in static benchmarks.

### 3.3. Game Variants

Here, we introduce four settings of *MetaNIM Arena*. Detailed explanation and theoretically determined winning strategies for these games are provided in Appendix A.

- **NIM:** Agents take turns removing 1 to $N$ (typically 3) objects from a set of heaps where the player who takes the last object wins. Success requires maintaining specific heap configurations to control the game's outcome.

- **Fibonacci:** A variation of NIM where an agent's maximum removal is constrained by the opponent's previous move. Each turn, an agent may remove between 1 and $2\times$ the opponent's last action. This rule introduces dynamic strategy adjustments, balancing immediate gains with long-term positioning.

- **Kayles:** Played with a or two row(s) of pins, where agents take turns knocking down one or two adjacent pins. The player unable to make a move loses. The challenge lies in evaluating pin configurations and predicting the opponent's responses to optimize each turn.

- **Chomp:** Played on a quadrangle grid, agents take turns consuming a "block" of chocolate along with all blocks below and to the right. The player forced to eat the top-right (or top-left) "poisoned" block loses.

- **Corner Queen:** On a rectangular board, two players take turns moving a queen left, down, or diagonally down-left. The first to reach the bottom-left corner wins. Strategy lies in limiting the opponent's options while advancing.

*Table 2.* Bias reinforcement across models: showing that even after the debate concludes, strongly consistent actions continue to exhibit strong consistency, reinforcing biased action distributions in the Fibonacci game. A wrong bias occurs when the model's biased response deviates from the optimal action, while a good bias refers to cases where the biased response aligns with the optimal action. (*a, b*) indicate the state where the remaining items are *a*, and player can take the items maximum to *b*.

| GPT-4o | Wrong Bias | | | Good Bias | | |
|---|---|---|---|---|---|---|
| | (20, 19) | (12, 4) | (7, 4) | (15, 10) | (16, 8) | (7, 7) |
| Standard | 0.700 | 0.675 | 0.600 | 0.525 | 0.725 | 0.850 |
| + After MAD | **0.900** | **0.750** | **0.700** | **0.750** | 0.750 | **0.900** |

| GPT-4o-mini | Wrong Bias | | | Good Bias | | |
|---|---|---|---|---|---|---|
| | (18, 4) | (12, 6) | (10, 4) | (15, 10) | (7, 2) | (15, 2) |
| Standard | 0.700 | 0.875 | 0.600 | 0.975 | 0.950 | 0.975 |
| + After MAD | **0.850** | **0.950** | **0.750** | **1.000** | 0.950 | **1.000** |

| GEMINI-1.5-pro | Wrong Bias | | | Good Bias | | |
|---|---|---|---|---|---|---|
| | (20, 19) | (12, 4) | (4, 4) | (15, 10) | (10, 4) | (16, 8) |
| Standard | 0.650 | 0.600 | 0.600 | 0.800 | 0.625 | 0.675 |
| + After MAD | **0.700** | **0.650** | **0.800** | 0.800 | **0.800** | **0.700** |

| GEMINI-1.5-flash | Wrong Bias | | | Good Bias | | |
|---|---|---|---|---|---|---|
| | (12, 6) | (12, 4) | (7, 4) | (15, 4) | (20, 19) | (7, 7) |
| Standard | 0.800 | 0.700 | 0.525 | 0.750 | 0.500 | 0.750 |
| + After MAD | **0.850** | **1.000** | **0.750** | 0.750 | **0.650** | **1.000** |

## 3.4. Combinatorial Games: Theory and Strategy

All *MetaNIM Arena* games are impartial combinatorial games, forming a Directed Acyclic Graph (DAG) where vertices represent game states and edges denote valid moves. The Grundy Number framework, along with the Sprague-Grundy Theorem, guarantees the existence of a winning strategy and provides a concrete method to determine it. In the *MetaNIM Arena* dataset, each state has a mathematically provable optimal move, allowing an LLM's decisions to be evaluated against the *theoretical optimal strategy*—a key advantage for unbiased assessment.

**Definition 3.1** (Grundy Number). For a finite impartial combinatorial game under normal play (where the last player to make a valid move wins), the *Grundy number* (or *Nimber*) $G(S)$ is recursively defined as follows. If $S$ is a *terminal state* with no valid moves, set $G(S) = 0$. Otherwise,

$$G(S) = \text{mex}\{ G(S') \mid S' \text{ is reachable from } S \}.$$

Here, $\text{mex}(X)$ is the smallest nonnegative integer not in $X$.

Note that the Grundy number is well-defined for every impartial combinatorial game, since the game's state space forms a DAG. In many combinatorial games, direct enumeration of all possible move sequences is computationally infeasible. However, with Sprague-Grundy Theorem, we can easily calculate Grundy Numbers on complex games. See Appendix A.1.1 for further discussions.

**Optimal Strategy.** When $G(S) \neq 0$, there is *at least one* move to a successor $S'$ with $G(S') = 0$, forcing the opponent into a losing position. Conversely, if $G(S) = 0$, *all* successor states have $G(S') \neq 0$. Because the game DAG is finite and acyclic, repeatedly applying "move to $G = 0$" (or avoiding it) ensures a *forced* result under optimal play. See Appendix A.1.2 for more details, including the *misère* variant where taking the last object *loses*.

# 4. Understanding Bias Reinforcement in Debate Process

To quantitatively analyze bias in the debate process, we define *strong consistency* and *bias reinforcement* as below.

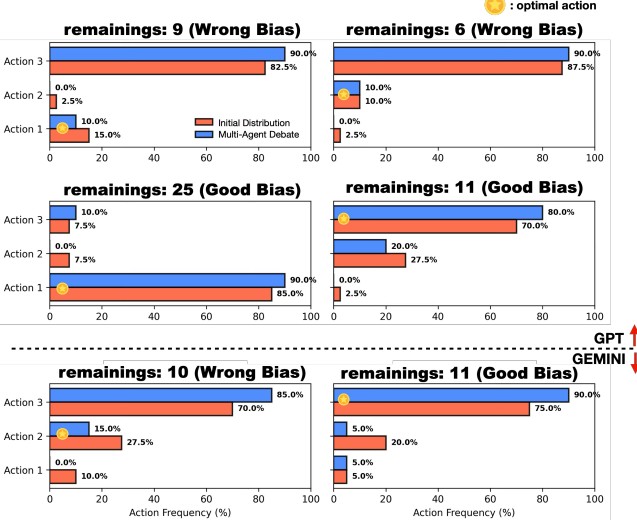

*Figure 2.* Bias reinforcement in NIM game by MAD. We compared initial action distribution and action distribution after 3 rounds of debates. MAD amplifies a model's biases, making debates favor consistent but potentially incorrect responses.

**Definition 4.1** (**Strong Consistency**). *Strong consistency* is a model's tendency to consistently produce the same output with high probability when given identical inputs. If a response's probability exceeds a threshold (set at 0.5) across multiple trials, we label the behavior as strongly consistent.

This phenomenon naturally emerges in strategic decision-making contexts. As shown in Figure 1, when presented with a specific game state, the model repeatedly generates the same reasoning pattern, highlighted in orange. Rather than effectively utilizing strategic prior knowledge, the model fixates on a single line of reasoning, limiting adaptability and decision quality.

**Definition 4.2** (**Bias Reinforcement**). *Bias reinforcement* in the context of large language models refers to the phenomenon where iterative reasoning processes—such as multi-agent debates—amplify pre-existing model biases instead of mitigating them. Rather than converging toward

a more accurate or optimal reasoning outcome, the debate process reinforces strongly consistent, yet potentially suboptimal or distorted, reasoning patterns.

In the rest of this section, we analyze bias reinforcement and lack of diversity in the debate process, using `GPT-4o-mini` and `GEMINI-1.5-pro` for the NIM game, and then extend the evaluation to Fibonacci with those two plus `GPT-4o` and `GEMINI-1.5-flash`.

### 4.1. Bias Reinforcement in MAD

**In NIM: (Figure 2)** We find that MAD amplifies models' pre-existing biases rather than refining their reasoning. To investigate this, we first identify game states where each model exhibits *strong consistency*, i.e., consistently selecting the same action across multiple trials. For each such state, we conduct multi-agent debates using two identical agents instantiated from the same model.

Each agent generates 20 responses (40 per state) at a fixed temperature of 0.7, maintained throughout the debate. Initial action distributions (light red) are compared against post-debate distributions after three rounds (blue). If MAD functioned as a self-correction mechanism, we would expect the distribution to shift toward the optimal action.

However, we found the opposite: regardless of whether the initial reasoning was correct, the debate process consistently amplifies pre-existing biases rather than mitigating them. For instance, in Figure 2 (top-left), `GPT-4o-mini` initially selects a suboptimal action (Action 3) 82.5% of the time. After the debate, this frequency increases to 90.0%, while the proportion of optimal responses drops further. Rather than correcting errors, the debate reinforces strongly consistent—but incorrect—responses. This pattern persists across model families. The GEMINI models (bottom row of Figure 2) exhibit similar behavior, regardless of whether the initial bias aligns with optimal play ("good bias") or not ("wrong bias"). In both cases, MAD strengthens the dominant trajectory without introducing new strategic insight.

Figure 1 provides a detailed view: two LLM agents receive the same input and begin debating. Despite initial divergence, they converge quickly—after the first round—on a shared line of reasoning. Crucially, this convergence occurs even when the initial consensus is incorrect, illustrating that MAD often serves as an amplifier of bias rather than a correction mechanism.

**In Fibonacci: (Table 2)** Extending our NIM analysis, we evaluate MAD's effect in the Fibonacci game, a more complex setting with move constraints and dynamic interactions. As in NIM, We identify states exhibiting *strong consistency* and categorize them into two groups: those where consistent responses align with the optimal strategy, and those where they do not. We then apply MAD to examine how response

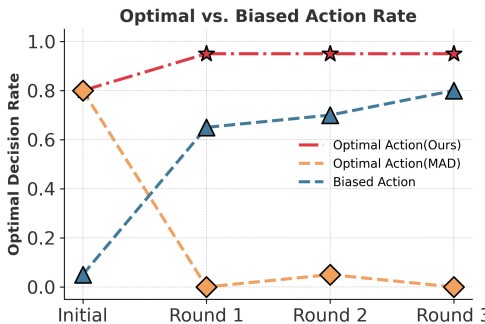

*Figure 3.* Decline in optimal actions over debate rounds, demonstrating the convergence toward consistently biased reasoning.

distributions evolve post-debate.

Consistent with the NIM results, MAD reinforces the model's initial biases in Fibonacci as well. As shown in Table 2, the frequency of initially consistent actions increases after debate, regardless of whether those actions are optimal. On average, reinforcement rises by 9.17% in GPT models and 12.29% in GEMINI models, indicating a systematic amplification of dominant reasoning patterns across architectures.

### 4.2. Optimal Inputs Do Not Reduce Bias in MAD

To further assess the bias reinforcement in MAD, we conduct an experiment on the NIM game (5 items remaining) using `GPT-4o-mini`, illustrated in Figure 3. We prepare two sets of 20 responses: one exhibiting *strong consistency* toward a biased action (Action 2), and another consisting of *optimal responses*, where 80% of actions correspond to the game-theoretic best move (Action 1). We then introduced these responses into a multi-agent debate setting using `GPT-4o-mini`, pairing each strongly consistent response with an optimal response, and observed how the model's reasoning evolves over multiple rounds of debate (denoted the game situation in detail in Appendix E).

Surprisingly, the debate fails to leverage the high-quality input. Initially, the curated dataset contained 80% optimal responses, yet after a single round of debate, the model predominantly aligned with the biased responses. As debate rounds proceed, this effect intensifies: the influence of the optimal input diminishes, while the initially frequent—but suboptimal—choice becomes dominant. These results demonstrate that MAD is not only ineffective at correcting bias, but can systematically aligns with its preexisting consistency patterns, reinforcing suboptimal but frequent choices.

In contrast, applying **DReaMAD** with the curated optimal responses preserves correct reasoning and mitigating bias reinforcement. This underscores the need for mechanisms that introduce diverse perspectives beyond internal debate, which we further discuss in §5.

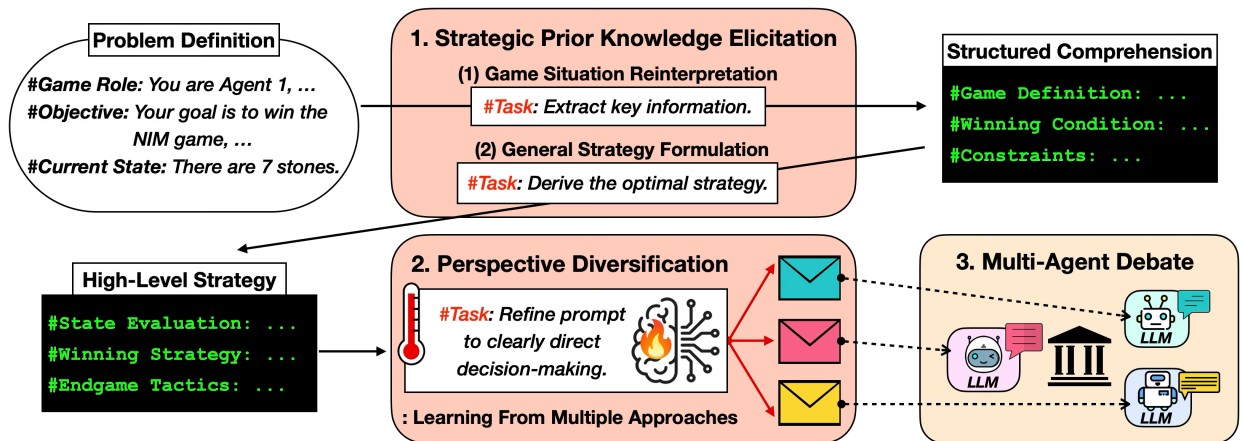

Figure 4. **DReaMAD** framework. **DReaMAD** improves LLM reasoning by combining **Strategic Prior Knowledge Elicitation** and **Perspective Diversification**. In the first stage, the model reinterprets the problem and formulates high-level strategies to reduce bias. In the second stage, multiple agents adopt distinct viewpoints, engage in structured debate, and refine their conclusions to enhance decision-making.

### 4.3. Lack of Reasoning Diversity in MAD

While MAD is designed to refine reasoning through agent interaction, its effectiveness is fundamentally constrained by a lack of genuine diversity (§2.3). To address this limitation, Chen et al. (2023b) propose a multi-model debate framework that combines outputs from different models to encourage diverse reasoning. In contrast, our approach focuses on single-model self-correction.

Interestingly, we observe that even within a single model, small variations in prompts can induce markedly different reasoning paths (Appendix D). For example, including or omitting the word *Fibonacci* leads to distinct strategies in the same task. This suggests that diversity in reasoning can be enhanced by strategically modifying prompts within the same model, providing a practical alternative to multi-model debate frameworks.

### 5. **DReaMAD**: Diverse Reasoning via MAD

To address the limitations of Multi-Agent Debate (MAD) and improve strategic decision-making in Large Language Models (LLMs), we introduce **DReaMAD** (**D**iverse **Rea**soning via **M**ulti-**A**gent **D**ebate with Refined Prompt). Our framework refines prior knowledge and ensures diverse perspectives by extending MAD in two main stages:

1. **Strategic Prior Knowledge Elicitation:** The model re-defines the problem, extracts key strategic insights, and formulates a high-level strategy before reasoning.

2. **Perspective Diversification:** Multiple agents are instantiated with self-generated distinct viewpoints to engage in dialectical reasoning.

After these stages, the agents conduct a structured multi-agent debate. A final post-debate refinement step then revisits their conclusions to improve reasoning quality. The complete workflow is illustrated in Figure 4 and the de-

tailed prompt formulation used in these two modules is well documented in the Appendix C.

### 5.1. Strategic Prior Knowledge Elicitation

To address strongly consistent bias, **DReaMAD** integrates a structured module that ensures systematic extraction and refinement of the LLM's internal strategic knowledge before the debate. First, the model is prompted to reinterpret the given problem, leading to a more organized understanding of the strategic context (**Game Situation Reinterpretation**). Next, it formulates a set of high-level strategies that can be applied to the scenario at hand (**General Strategy Formulation**), preventing the model from settling too early on potentially flawed reasoning. As shown in Table 3, this module enhances the model's performance in tasks that require strategic prior knowledge (Figure 4-1). We set the *temperature* hyperparameter to be 0.1 for strategic consistency.

### 5.2. Perspective Diversification

Building on MAD, **DReaMAD** mitigates argument homogenization by ensuring each agent adopts a distinct viewpoint prior to the debate. This approach is inspired by the *Learning from Multiple Approaches* concept in education theory (Council et al., 2005; Cleaves, 2008), which suggests that individuals improve their problem-solving skills by exploring multiple representations of the same problem. Analogously, this module ensures that each agent is given differentiated initial prompts that guide reasoning along distinct strategic trajectories. By self-customizing initial prompts for each model instance, **DReaMAD** encourages unique strategic perspectives and reduces the risk of bias reinforcement (Figure 4-2) as demonstrated in Figure 5 and Figure 6 (left). In Figure 5, although each agent receives the same initial prompt, they independently generate different optimal prompts, leading to distinct distributions of reasoning. This process reduces each agent's bias and fosters a more robust debate. Remarkably, even in a state with very

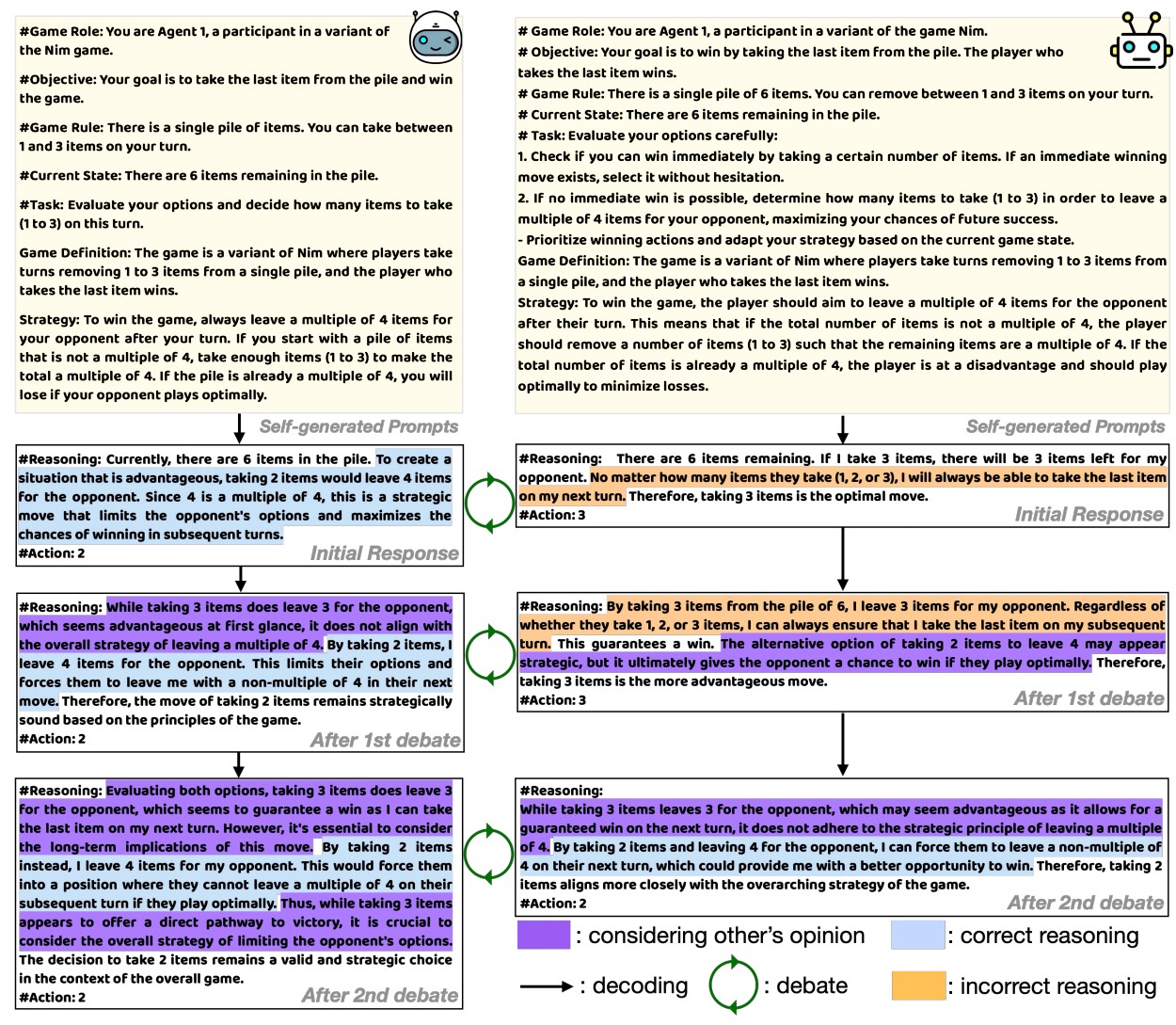

Figure 5. In this example, we illustrate how the debate process converges to an optimal outcome using **DReaMAD**. We begin with the same *current state* shown in Figure 1, employing self-generated prompts for each LLM agent.

strong consistency, the discussion converges well toward the correct reasoning direction. Here, we set *temperature* to be 0.7 for diversity.

## 6. Experiments

We utilized the benchmark *MetaNIM Arena* as both our dataset and simulator, as it provides a controlled environment for evaluating reasoning under grounded strategic tasks. Our investigation focuses on three key questions: (1) Does our approach improve reasoning quality compared to existing prompting techniques? (2) Does our approach prove its strategic reasoning quality under adversarial decision-making environments? (3) Does generating diverse prompts contribute to better decision-making within the debate framework?

To address these questions, we compare **DReaMAD** with standard prompting methods including ReAct (Yao et al.,

2023), Chain-of-Thought (CoT), Self-Consistency (Wang et al., 2022), Self-Refinement (Madaan et al., 2024), and Multi-Agent Debate (MAD (Du et al., 2023), MAD2 (Liang et al., 2023)). The details of standard and CoT prompts are provided in Appendix B.

Our method builds on the MAD framework by Du et al. (2023), augmenting it with structured self-prompt refinement and perspective diversification. For self-refinement, we follow the methodology of Madaan et al. (2024), applying three iterative refinement steps. Similarly, for MAD, we conducted up to three rounds of debate, following Du et al. (2023), with the process terminating early if a consensus is reached before the final round.

We also investigate whether the observed improvements generalize across different LLM architectures, including both GPT and GEMINI models.

*Table 3.* Effect of Strategic Prior Knowledge Elicitation module. **DReaMAD**$^{(-)}$ indicates our method except multi-agent debate process. We can fully evaluate reasoning ability between different prompting methods. The metric accuracy of selecting optimal action is used. The best results are highlighted in **bold**.

| LLM Models | Prompting Methods | NIM | Fibonacci | Chomp | Kayles | Average |
|---|---|---|---|---|---|---|
| GPT-4o | ReAct | $0.95 \pm 0.04$ | $0.33 \pm 0.04$ | $0.18 \pm 0.07$ | $0.19 \pm 0.08$ | 0.41 |
| | + CoT-Prompting | $0.96 \pm 0.04$ | $0.43 \pm 0.11$ | **$0.28 \pm 0.09$** | $0.20 \pm 0.10$ | 0.47 |
| | **DReaMAD**$^{(-)}$ | **$0.98 \pm 0.04$** | **$0.44 \pm 0.09$** | $0.23 \pm 0.10$ | **$0.23 \pm 0.12$** | 0.47 |
| GPT-4o-mini | ReAct | $0.75 \pm 0.05$ | $0.33 \pm 0.04$ | $0.40 \pm 0.07$ | $0.12 \pm 0.06$ | 0.40 |
| | + CoT-Prompting | $0.84 \pm 0.08$ | $0.36 \pm 0.04$ | $0.61 \pm 0.05$ | $0.02 \pm 0.03$ | 0.46 |
| | **DReaMAD**$^{(-)}$ | **$1.00 \pm 0.00$** | **$0.49 \pm 0.17$** | **$0.62 \pm 0.10$** | **$0.18 \pm 0.11$** | **0.57** |
| GEMINI-1.5-pro | ReAct | $0.82 \pm 0.06$ | $0.42 \pm 0.04$ | $0.19 \pm 0.08$ | $0.57 \pm 0.04$ | 0.50 |
| | + CoT-Prompting | $0.88 \pm 0.05$ | $0.47 \pm 0.11$ | $0.22 \pm 0.11$ | $0.59 \pm 0.04$ | 0.54 |
| | **DReaMAD**$^{(-)}$ | **$0.97 \pm 0.04$** | **$0.53 \pm 0.07$** | **$0.24 \pm 0.05$** | **$0.72 \pm 0.12$** | **0.62** |
| GEMINI-1.5-flash | ReAct | $0.94 \pm 0.02$ | $0.35 \pm 0.04$ | $0.05 \pm 0.03$ | $0.01 \pm 0.02$ | 0.34 |
| | + CoT-Prompting | $0.93 \pm 0.02$ | $0.33 \pm 0.07$ | **$0.09 \pm 0.04$** | $0.0 \pm 0.00$ | 0.34 |
| | **DReaMAD**$^{(-)}$ | **$0.97 \pm 0.04$** | **$0.45 \pm 0.06$** | $0.05 \pm 0.06$ | **$0.42 \pm 0.06$** | **0.46** |
| **Average** | ReAct | 0.87 | 0.36 | 0.21 | 0.22 | - |
| | + CoT-Prompting | 0.90 | 0.40 | **0.30** | 0.20 | - |
| | **DReaMAD**$^{(-)}$ | **0.98** | **0.48** | 0.29 | **0.39** | - |

*Table 4.* Winning rate comparison across different models and different self-correction methods. This is an result based on *MetaNIM Arena* simulator. The best results are highlighted in **bold**.

| LLM Models | Prompting Methods | NIM | | Fibonacci | | Kayles | | Chomp | | C.Queen |
|---|---|---|---|---|---|---|---|---|---|---|
| | | Normal | Misère | Normal | Misère | Single | 2 Rows | Rectangular | Square | Normal |
| GEMINI-1.5-flash | Standard Prompting | 0.32 | 0.54 | 0.16 | 0.10 | 0.54 | 0.56 | **0.78** | 0.18 | 0.46 |
| | + ReAct | 0.10 | 0.68 | 0.16 | 0.76 | 0.50 | 0.30 | 0.42 | 0.12 | 0.46 |
| | + Self-Refinement | 0.14 | 0.66 | 0.18 | 0.36 | 0.50 | 0.46 | 0.46 | 0.16 | 0.42 |
| | + Self-Consistency | 0.04 | 0.28 | **0.28** | 0.86 | 0.30 | 0.24 | 0.74 | 0.0 | 0.34 |
| | + MAD | 0.06 | 0.30 | 0.12 | 0.78 | 0.54 | 0.20 | 0.74 | 0.14 | 0.58 |
| | + MAD2 | 0.26 | 0.26 | 0.08 | 0.06 | 0.44 | 0.36 | 0.68 | 0.10 | 0.58 |
| | **+ DReaMAD** | **0.38** | **0.84** | 0.16 | **0.94** | **0.58** | **0.62** | 0.60 | **0.22** | **0.74** |
| GPT-4o-mini | Standard Prompting | 0.38 | 0.54 | 0.22 | 0.28 | 0.46 | 0.48 | 0.46 | 0.38 | 0.12 |
| | + ReAct | 0.22 | 0.68 | 0.20 | 0.34 | 0.40 | 0.26 | 0.58 | 0.34 | 0.28 |
| | + Self-Refinement | 0.22 | 0.70 | 0.18 | 0.50 | 0.46 | 0.52 | 0.52 | 0.44 | 0.24 |
| | + Self-Consistency | 0.14 | 0.52 | 0.34 | 0.46 | 0.32 | 0.20 | 0.54 | 0.26 | 0.30 |
| | + MAD | 0.28 | 0.62 | 0.22 | 0.64 | 0.42 | 0.28 | 0.52 | 0.56 | 0.44 |
| | + MAD2 | 0.34 | 0.20 | 0.18 | 0.18 | 0.50 | 0.34 | 0.44 | **0.90** | 0.14 |
| | **+ DReaMAD** | **0.98** | **0.74** | **0.54** | **0.72** | **0.68** | **0.84** | **0.64** | 0.22 | **0.76** |

## 6.1. Does **DReaMAD** Improve Reasoning Quality?

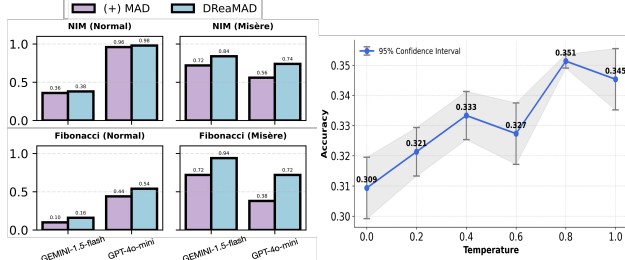

*Figure 6.* Effect of perspective diversification. Left: Average win rate of (+)MAD and **DReaMAD** on NIM and Fibonacci (Normal and Misère variants), aggregated over 50 simulations per setting. Right: Accuracy on the Fibonacci benchmark with GPT-4o across different sampling temperatures (15 runs each, 95% CI). Higher temperatures yield greater prompt diversity, leading to improved accuracy.

This experiment isolates the effect of strategic prior knowledge elicitation (§5.1), allowing us to assess whether our method enhances decision-making without relying on debate dynamics.

**Setup.** To evaluate the effectiveness of our approach in improving reasoning capabilities, we compare **DReaMAD** without the debate process against ReAct and zero-shot CoT prompting across multiple models in the *MetaNIM Arena* dataset (§A.3). For showing versatility of **DReaMAD**, we conduct experiments on four variants of LLMs as shown in Table 3.

**Results.** Table 3 demonstrates that **DReaMAD** consistently outperforms both ReAct and CoT prompting across all models and tasks. These results highlight the impact of our method in reinforcing structured strategic reasoning, even without the iterative correction process of debate. Notably, our approach leads to substantial improvements in NIM, Fibonacci, and Kayles, which are environments where long-term strategic planning plays a crucial role. Since defining a general winning strategy in Chomp is non-trivial, applying prior knowledge is challenging and results in less effectiveness compared to other games. Furthermore, we observe that models with inherently weaker reasoning abilities benefit the most from strategic prior knowledge elicitation (*e.g.,* GPT-4o-mini with +17%p, GEMINI-1.5-flash with +12%p on average).

*Table 5.* Accuracy (%) on math-reasoning benchmarks. Columns group the five algorithms for each backbone model. Bold indicates the best algorithm for a given (dataset, model) pair.

| Dataset | GPT-o3-mini | | | | | GPT-4o | | | | |
|---|---|---|---|---|---|---|---|---|---|---|
| | ReAct | Self-Refinement | Self-Consistency | MAD | **DReaMAD** | ReAct | Self-Refinement | Self-Consistency | MAD | **DReaMAD** |
| AIME 2024 | 76.7 | 73.3 | 86.7 | 73.3 | **90.0** | 0.0 | 3.3 | **10.0** | 3.3 | **10.0** |
| AMC 2023 | 97.5 | **100** | **100** | **100** | **100** | 60.0 | 52.5 | 52.5 | 60.0 | **62.5** |

*Table 6.* Accuracy (%) on CommonsenseQA dataset. **Bold** indicates the best performance. We abbreviate Self-Refinement as Self-Refine. and Self-Consistency as Self-Consist.

| Model | ReAct | Self-Refine. | Self-Consist. | MAD | **DReaMAD** |
|---|---|---|---|---|---|
| GPT-4o | 83.6 | 49.2 | 83.6 | 82.0 | **84.4** |
| GPT-4o-mini | 78.7 | 61.5 | 78.7 | 75.4 | **79.5** |

## 6.2. **DReaMAD** in Adversarial Strategic Decision-Making

**Setup.** We applied **DReaMAD** to GPT-4o-mini and GEMINI-1.5-flash and compared it with self-correction methods, including standard-prompt, ReAct, self-refinement, self-consistency, and MAD. To demonstrate its effectiveness, we used GPT-4o as the opponent model due to its superior performance. In this experiments, we utilized *MetaNIM Arena* simulator (§A.4) to maximize the effect of generating diverse prompts. We aimed to validate our hypothesis in a simulator that requires strategic decision-making within complex dynamics. We ran 50 independent episodes and average the win-rate.

**Results.** As shown in Table 4, **DReaMAD** consistently outperforms other self-correction methods across various strategic environments, demonstrating a significant improvement in winning rates. This result suggests that our approach enables LLM agents to effectively adapt to complex dynamics, particularly in adversarial decision-making scenarios where strategic reasoning is crucial. However, we observe that **DReaMAD** struggles in the Chomp game, which aligns with our hypothesis that Chomp lacks a well-defined generalized winning strategy. Unlike other tested environments, Chomp requires more exploratory play rather than direct reasoning from prior knowledge, highlighting a limitation of our method in environments where strategic heuristics are less structured.

## 6.3. Does Generating Diverse Prompts Improve Performance?

We assess whether prompt diversity improves decision quality within the MAD framework. To this end, we compare two settings: (1) identical prompts generated via the Strategic Prior Knowledge Elicitation module for both agents ((+)MAD), and (2) distinct, self-generated prompts per agent, as in **DReaMAD** (Figure 6, left). Experiments were conducted on four variants of the *MetaNIM Arena* simulator, NIM (Normal and Misère) and Fibonacci (Normal and Misère), over 50 episodes each (§A.4). Our results show that

incorporating diverse prompts within the MAD framework significantly enhances performance, validating the effectiveness of our Perspective Diversification module.

We also examine the effect of sampling temperature on prompt diversity in the Fibonacci task. Within both the Strategic Prior Knowledge Elicitation and Perspective Diversification modules, we vary the temperature from 0.0 to 1.0. As shown in Figure 6 right, higher temperature (further diversity) correlates with increased optimal action accuracy, indicating that greater diversity in generated prompts contributes to improved reasoning performance.

## 6.4. Generalization to Math and Commonsense Reasoning

While our primary benchmark focuses on structured games, we further evaluate **DReaMAD** on NLP tasks to test its broader applicability. Specifically, we consider algebra and number theory problems from AIME 2024 and AMC 2023, as well as CommonsenseQA (Talmor et al., 2018), which require symbolic reasoning and multi-step inference. These tasks are representative of domains where chain-of-thought reasoning is essential, making them suitable for evaluating generalization. We also examine whether **DReaMAD** benefits reasoning-specialized models, such as GPT-o3-mini, in a similar manner as general-purpose LMs. Experiments are conducted using standard accuracy metrics across model-task pairs. Results in Table 5 and Table 6 show that **DReaMAD** not only boosts accuracy across heterogeneous tasks but also yields clear gains on reasoning-oriented models such as GPT-o3-mini, underscoring the method's robustness well beyond the structured-game domain.

## 7. Conclusions

Our study shows that Multi-Agent Debate (MAD) often reinforces biases instead of reducing them, leading to suboptimal reasoning. Through our experiments with the *MetaNIM Arena*, we have observed that models persist in biased reasoning even when presented with superior alternatives. While our current strategy focuses on strategic games, the principles of structured self-refinement and diversified reasoning could be valuable for a wider range of NLP tasks. These include complex activities such as multi-step reasoning in question answering, legal analysis, and scientific inference. Future work will explore how these techniques enhance decision-making beyond structured games.

## Acknowledgments

This work was supported by Center for Applied Research in Artificial Intelligence (CARAI) grant funded by DefenseAcquisition Program Administration (DAPA) and Agency for Defense Development (ADD) (UD230017TD).

## Impact Statement

This work advances Machine Learning by enhancing LLMs' strategic reasoning through our approach, addressing bias reinforcement and lack of perspective diversity in Multi-Agent Debate (MAD). Our method improves decision-making in adversarial settings, with potential applications in automated negotiations, economics, and multi-agent systems. However, stronger AI-driven strategies could be misused in manipulative or deceptive contexts. To ensure ethical deployment, future research should focus on integrating fairness constraints and transparency mechanisms in AI decision-making.

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

- Appendix -

# Understanding Bias Reinforcement in LLM Agents Debate

## A. MetaNIM Arena

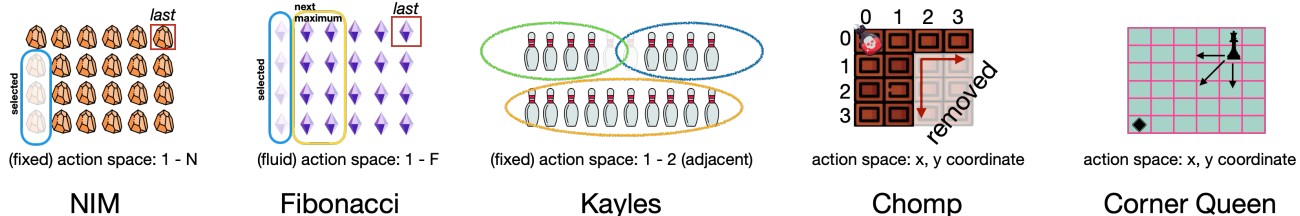

*Figure 7.* The concept illustration of MetaNIM Arena.

---

**Algorithm 1 MetaNIM Arena:** Turn-Based Opponent Task with Two Agents

---

**Require:** Initial State $S_0$, Goal Condition $G$, Agents $A_1$ and $A_2$
**Ensure:** Final State $S_f$ and Outcome

1:   $t \leftarrow 0$         *Initialize turn counter*
2:   $S \leftarrow S_0$         *Set initial state*
3:  **while** $S \notin G$ **and** game is not terminated **do**
4:     **if** $t \bmod 2 = 0$ **then**
5:       $a_t \leftarrow A_1(S)$     *Agent 1's turn, selects action $a_t$*
6:     **else**
7:       $a_t \leftarrow A_2(S)$     *Agent 2's turn, selects action $a_t$*
8:     **end if**
9:     $S \leftarrow \text{UpdateState}(S, a_t)$   *Apply the action and update state*
10:    $t \leftarrow t + 1$     *Increment turn counter*
11:  **if** $S \in G$ **then**
12:     **Success**: Goal Achieved
13:  **end if**
14: **end while**

---

### A.1. Theoretical Background on Combinatorial Impartial Games

Each game in MetaNIM Arena is a combinatorial impartial game. We begin by outlining the relevant theoretical foundation, starting with the Sprague–Grundy theorem.

#### A.1.1. SPRAGUE-GRUNDY THEOREM

The Sprague-Grundy theorem provides a fundamental method for analyzing impartial combinatorial games by decomposing complex games into simpler, independent subgames. As discussed in Section 3.4, every impartial combinatorial game can be represented as a directed acyclic graph (DAG). However, directly computing Grundy numbers recursively from terminal states is often impractical.

We summarize key results from Sprague (1935) and Grundy (1939). According to the theorem, the optimal strategy for playing multiple impartial games simultaneously (in parallel), or a single complex game viewed as multiple independent subgames, is equivalent to playing a single game of Nim with multiple heaps. This equivalence arises from the concept of the disjunctive sum of DAGs.

**Definition A.1** (Disjunctive Sum of DAGs). Let $\mathcal{G}_1 = (X_1, F_1), \mathcal{G}_2 = (X_2, F_2), \ldots, \mathcal{G}_n = (X_n, F_n)$ be DAGs representing

$n$ impartial combinatorial games. The disjunctive sum of $\mathcal{G}_1, \ldots, \mathcal{G}_n$ is a DAG $\mathcal{G} = (X, F)$ defined as follows:

1. The vertex set $X$ is the Cartesian product $X_1 \times X_2 \times \cdots \times X_n$.

2. The edge set $F$ consists of edges connecting $(x_1, \ldots, x_n)$ to $(y_1, \ldots, y_n)$ if and only if exactly one pair $(x_i, y_i)$ is in $F_i$, and $x_j = y_j$ for all $j \neq i$.

**Note.** In a disjunctive sum of DAGs, each player chooses exactly one subgame to play during their turn and moves within that subgame. The entire game ends when all subgames reach terminal positions.

**Theorem A.2** (Sprague-Grundy (Sprague, 1935; Grundy, 1939)). *A position $S$ is losing if and only if its Grundy number $G(S) = 0$; otherwise, if $G(S) \neq 0$, it is winning. Furthermore, if a position $S$ decomposes into independent subpositions $S_1, \ldots, S_k$ via the disjunctive sum of DAGs, then*

$$G(S)_{(2)} = G(S_1)_{(2)} \oplus G(S_2)_{(2)} \oplus \cdots \oplus G(S_k)_{(2)},$$

*where $\oplus$ denotes bitwise XOR.*

This result implies that Grundy numbers for complex games, such as Kayles or Chomp, can be efficiently computed by decomposing them into simpler subgames and combining the Grundy numbers using bitwise XOR.

For example, consider a variant of the game Kayles played on two separate rows of pins, each forming an independent subgame. Suppose we computed the Grundy numbers separately for these rows, obtaining Grundy numbers $7$ for the first row and $4$ for the second row. By the Sprague-Grundy theorem, the combined Grundy number of the position is given by:

$$7_{(2)} \oplus 4_{(2)} = 111_{(2)} \oplus 100_{(2)} = 011_{(2)} = 3.$$

Thus, even though the original game involves two distinct rows of pins, the strategic analysis reduces precisely to analyzing a Nim heap of size 3. Since a Nim heap of size 3 is nonzero, this indicates a winning position for the player about to move.

A.1.2. BASIC DISCUSSIONS ON THE OPTIMAL STRATEGY

**Why $G(S) = 0$ Implies Losing.** Recall that $G(S)$ is defined as:

$$G(S) = \text{mex}\left\{ G(S') \,\middle|\, S' \text{ is reachable from } S \right\},$$

where $\text{mex}(X)$ is the smallest nonnegative integer *not* in the set $X$. Thus,

$$G(S) = 0 \iff 0 \notin \left\{ G(S') : S' \text{ is reachable from } S \right\}.$$

Concretely, if $G(S) = 0$, then *no valid move* leads to a successor $S'$ with $G(S') = 0$. In other words, from $S$, the player to move *cannot* transition the game into a $G(\cdot) = 0$ state. Because a state $G(S') = 0$ corresponds to a losing position *for the player who faces it*, the mover in state $S$ has *no way* to force the opponent into a losing position on the next turn. Hence, $S$ is losing for the player to move.

**Why $G(S) \neq 0$ Implies Winning (Opposite viewpoint).** By the same logic, if $G(S) \neq 0$, then the definition of $\text{mex}$ guarantees 0 *does* appear among the Grundy values $G(S')$ of the successors. Thus, there *exists* some child state $S'$ for which $G(S') = 0$. Consequently, the current mover can place the opponent directly into a losing position (i.e. a position with Grundy number 0). Recursively iterating this argument along the Directed Acyclic Graph of states ensures that the current mover, if playing optimally, keeps forcing the opponent into $G(\cdot) = 0$ states until the game ends. Therefore, $S$ must be a *winning* state.

**Misère Variant.** *Misère* play reverses the normal condition: taking the last object loses rather than wins. Although standard Sprague-Grundy analysis still applies to most states, a special exception arises when all heaps (or subpositions) are size 1, such as in *misère* Nim. In that endgame scenario, the usual strategy must switch to avoid forcing the final move, ensuring the player leaves the opponent to pick the last object.

## A.2. Game Variants in MetaNIM Arena

### A.2.1. NIM

Nim is a mathematical strategy game where two players alternate turns removing objects from distinct heaps/piles. The classic version follows these rules:

- **Heaps**: The game starts with $k$ heaps containing $n_1, n_2, \ldots, n_k$ objects respectively

- **Moves**: On their turn, a player must remove at least 1 to previously fixed number of objects from exactly one heap

- **Objective**: The player who takes the last remaining object wins (normal play convention)

**Mathematical Strategy**    The game can be analyzed using binary representations through the concept of **Nimbers** (Grundy numbers). For any position, the key is to calculate the binary XOR (exclusive OR) sum of all heap sizes:

$$\text{Nim-sum} = n_1 \oplus n_2 \oplus \cdots \oplus n_k$$

A position is **losing** if the Nim-sum equals 0. The winning strategy consists of always moving to a position with Nim-sum 0. For the single-heap variant (as in our current game), this simplifies to maintaining modular arithmetic conditions.

**Example**    Consider a game with heaps [3, 4, 5]:

$$3 = 011_2$$
$$4 = 100_2$$
$$5 = 101_2$$
$$\text{Nim-sum} = 011_2 \oplus 100_2 \oplus 101_2 = 010_2 = 2 \neq 0$$

The first player can win by removing 2 objects from the 5-object heap to make the new Nim-sum 0.

**Variants**    Several Nim variants exist, including:

- Single-heap Nim (as in our current game)

- Misère Nim (player taking last object *loses*)

- Multi-heap Nim with different removal constraints

The fundamental mathematical principles of combinatorial game theory apply to all variants.

### A.2.2. FIBONACCI

The Fibonacci Game, also known as **Fibonacci Nim**, is a combinatorial number game where players alternate removing items from a pile, with move constraints based on the Fibonacci sequence. The rules are:

- **Initial Move**: First player takes $1 \leq k < n$ items from a pile of $n$ items

- **Subsequent Moves**: Each player must take between 1 and *twice* the number of items taken by their opponent in the previous move

- **Objective**: The player who takes the last item wins

**Mathematical Strategy**   The game is governed by Fibonacci numbers ($F_1 = 1, F_2 = 2, F_n = F_{n-1} + F_{n-2}$) and **Zeckendorf's Theorem**, which states that every positive integer can be uniquely expressed as a sum of non-consecutive Fibonacci numbers.

- **Losing Positions**: Pile sizes equal to Fibonacci numbers ($F_n$)

- **Winning Strategy**: Reduce the pile to the largest Fibonacci number smaller than the current size

For a pile of size $m$, its Zeckendorf representation is:

$$m = F_{k_1} + F_{k_2} + \cdots + F_{k_r} \quad (|k_i - k_j| \geq 2)$$

The optimal first move is to remove the *smallest* Fibonacci number in this decomposition.

**Example**   For a starting pile of $m = 20$:

$$\text{Zeckendorf: } 20 = 13 + 5 + 2 \quad (F_7 = 13, F_5 = 5, F_3 = 2)$$
$$\text{First move} = \text{Remove smallest term } 2$$
$$\text{New pile} = 18 = 13 + 5$$

Now the opponent faces a position composed purely of Fibonacci numbers. Any move they make ($1 \leq x \leq 4$) can be countered by reducing the pile to the next Fibonacci number.

**Key Properties**

- If $m$ is a Fibonacci number, the first player will lose against perfect play

- The number of moves in a game is always $\leq$ the index of the largest Fibonacci number $\leq m$

- The golden ratio $\phi = \frac{1+\sqrt{5}}{2}$ emerges in win/loss probability analysis

**Variants**

- Reverse Fibonacci Nim (last player to move loses)

- Multi-pile Fibonacci games

- Constrained Fibonacci sequences (e.g., Tribonacci variants)

This game demonstrates deep connections between combinatorial game theory, number theory, and the Fibonacci sequence.

A.2.3. KAYLES

Kayles is an impartial combinatorial game played with a linear arrangement of pins where players alternate knocking down pins under specific adjacency rules. First analyzed in 1929 by Dudeney and later studied by Conway and Berlekamp, it demonstrates complex mathematical patterns.

**Basic Rules**

- **Initial Setup**: A row of $n$ identical pins

- **Moves**: On each turn, a player must either:
    - Knock down 1 pin
    - Knock down 2 adjacent pins

- **Objective**: Last player to make a valid move wins (normal play convention)

**Mathematical Analysis**    The game is analyzed using **Grundy numbers** and the **Sprague-Grundy theorem**. Positions split into independent segments after moves create disjunctive game components.

- Let $G(n)$ be the Grundy number for a row of $n$ pins

- Recursive Grundy number calculation:

$$G(n) = \text{mex}\{G(n-1), G(n-2), G(a) \oplus G(b)\}$$

  where $a + b = n - k$ for $k \in \{1, 2\}$, and $\text{mex}$ = minimum excludant

**Key Patterns**

- Positions with Grundy number 0 are losing positions

- The Grundy sequence becomes periodic with period 12 for large $n$

- Known solution: $G(n) = n \mod 12$ when $n \geq 70$

**Example**    Consider a row of 4 pins:

$$G(0) = 0$$
$$G(1) = \text{mex}\{G(0)\} = 1$$
$$G(2) = \text{mex}\{G(1), G(0), G(0) \oplus G(0)\} = \text{mex}\{1, 0, 0\} = 2$$
$$G(3) = \text{mex}\{G(2), G(1), G(1) \oplus G(0)\} = \text{mex}\{2, 1, 1\} = 0$$
$$G(4) = \text{mex}\{G(3), G(2), G(2) \oplus G(0)\} = \text{mex}\{0, 2, 2\} = 1$$

A row of 4 pins has Grundy number 1, making it a winning position.

**Strategic Principles**

- Split long rows into independent segments with XOR-sum 0

- Mirror opponent's moves in symmetric positions

- Avoid leaving isolated single pins

**Variants**

- Circular Kayles (pins arranged in a circle)

- Multi-row Kayles

- $k$-Kayles (allow knocking down up to $k$ adjacent pins)

- Misère Kayles (last player to move loses)

**Computational Complexity**    Kayles is:

- PSPACE-complete for general positions

- Solved in linear time for standard single-row play

- Used in complexity theory to study impartial games

This analysis demonstrates how simple rule sets can generate complex mathematical structures. The complete Grundy number sequence for Kayles was only fully determined through extensive computational analysis.

A.2.4. CHOMP

Chomp is an impartial combinatorial game first formulated by David Gale in 1974. Played on a rectangular grid representing a chocolate bar, it features unique topological constraints and demonstrates fundamental principles of partially ordered sets (posets).

**Basic Rules**

- **Initial Setup**: An $m \times n$ rectangular grid of "chocolate squares"

- **Special Square**: The lower-left square (position (1,1)) is poisoned

- **Moves**: On each turn, a player must:
  - Select any remaining square
  - Remove ("chomp") all squares above and/or to the right of the selected square

- **Objective**: Avoid taking the poisoned square - last player to make a valid move wins (normal play convention)

**Mathematical Analysis**    Chomp is particularly significant in combinatorial game theory because:

- It is a **partisan game** with inherent asymmetry

- The starting position is a poset under component-wise ordering

- A winning strategy exists for the first player (proven by strategy-stealing argument), though explicit strategies are unknown for most grid sizes

**Key Theorem (Gale, 1974)**

**Theorem A.3.** *For any initial grid size $m \times n$ where $m, n \geq 2$, the first player has a winning strategy.*

**Example: 2 × 3 Grid**

$$\begin{bmatrix} \circ & \circ & \circ \\ \bullet & \circ & \circ \end{bmatrix}$$

First player wins by taking the (2,3) square:

- If Player 2 takes (1,3), Player 1 takes (2,2)

- If Player 2 takes (2,2), Player 1 takes (1,2)

- All paths eventually force Player 2 to take the poison

**Strategic Principles**

- Maintain control of the antidiagonal

- Force symmetry when possible

- Reduce the game to independent subgames

- Avoid leaving isolated columns

**Computational Complexity**

- General Chomp is PSPACE-complete

- Solved in polynomial time for:
    - $2 \times n$ grids
    - Square grids up to $5 \times 5$

- Number of winning positions grows exponentially with grid size

**Variants**

- 3D Chomp (cuboidal grids)

- Circular Chomp

- Hypergraph Chomp

- Misère Chomp (taking poison square wins)

- Numerical Chomp (played on factor lattices)

**Significance**   Chomp demonstrates fundamental connections between:

- Combinatorial game theory

- Computational complexity

- Algebraic geometry (via Gröbner basis interpretations)

Despite its simple rules, Chomp remains unsolved for general grid sizes, making it an active research area in computational combinatorics.

A.2.5. CORNER QUEEN

Corner Queen is a two-player combinatorial impartial game played on a rectangular grid, where a single queen starts at an arbitrary position and players alternate moving it toward the bottom-left corner. The player who moves the queen to the target wins.

**Basic Rules**

- **Initial Setup**: A queen is placed on an $m \times n$ grid at position $(x, y)$

- **Moves**: On each turn, a player moves the queen in one of the following directions:
    - Left: $(x, y) \rightarrow (x - k, y)$
    - Down: $(x, y) \rightarrow (x, y - k)$
    - Diagonally down-left: $(x, y) \rightarrow (x - k, y - k)$
    - for any $k \in \mathbb{Z}_{>0}$ such that the move stays within the board

- **Objective**: The player who moves the queen to position $(0, 0)$ wins

**Mathematical Analysis**    Corner Queen is mathematically equivalent to **Wythoff's Game**, a well-studied impartial game in combinatorial game theory. Each game state $(x, y)$ corresponds to a position on the board where $x, y \in \mathbb{N}$ and $x \leq y$ (without loss of generality).

- The game's Grundy function $G(x, y)$ satisfies analyzed via Beatty sequences for Wythoff's Game

- The **losing positions** (also called *P-positions*) are given by pairs of the form:

$$(\lfloor k\phi \rfloor, \lfloor k\phi^2 \rfloor), \quad k \in \mathbb{N}$$

where $\phi = \frac{1+\sqrt{5}}{2}$ is the golden ratio

- **Examples:** The first few $P$-positions are $(1, 2)$, $(3, 5)$, $(4, 7)$, $(6, 10)$, $(8, 13)$, and $(9, 15)$.

**Optimal Strategy**

- A position is losing if and only if it lies on the Wythoff pairs described above

- The winning strategy is to move the queen to the nearest $P$-position

- These positions are sparse and non-periodic, but can be computed efficiently using Beatty sequences

### A.3. Our Dataset used in Experiments

We construct simple dataset based on the *MetaNIM Arena*. This dataset doesn't require any opponent model because samples in this dataset is focusing on the specific scene in each game.

*Table 7.* Constructed dataset using *MetaNIM Arena*. We evaluate models on this dataset and report results in Table 3.

| Methods | NIM | Fibonacci | Chomp | Kayles |
|---|---|---|---|---|
| Action space | 1–3 | dynamic (max 30) | $x, y$ coordinate (scenario-dependent) | single pin or two adjacent pins |
| Variants | Normal | Normal | Square (2×2 – 19×19) | Normal |
| # samples | 20 | 11 | 20 | 18 |

### A.4. Our Simulator used in Experiments

We build simulator based on the *MetaNIM Arena*. This simulator requires any opponent available to receive prompt and make an output as an action. Here, we utilize the gpt-4o model as an opponent.

| Features | NIM | | Fibonacci | |
|---|---|---|---|---|
| | Normal | Misère | Normal | Misère |
| **Starting Point** | remaining items: 31 | remaining items: 31 | remaining items: 20 | remaining items: 20 |
| **Winning Condition** | taking last item | avoiding last item | taking last item | avoiding last item |
| **First Player?** | ✓ | ✓ | ✓ | ✓ |
| **Action Space** | 1 - 3 | 1 - 3 | dynamic | dynamic |
| **Opponent** | `GPT-4o-2024-08-06` | | | |

*Table 8.* MetaNIM Arena Simulator: NIM and Fibonacci

| Features | Kayles | | Chomp | | Corner Queen |
|---|---|---|---|---|---|
| | Single | 2 Rows | Rectangular | Square | Normal |
| **Starting Point** | remaining items: 20 | piles 5 6 | 2×8 | 5×5 | queen at $(4, 16)$ |
| **Winning Condition** | take last item | take last item | avoid poison (top-left) | avoid poison (top-right) | reach $(0, 0)$ (lower-left corner) |
| **First Player?** | ✓ | ✓ | ✓ | ✓ | ✓ |
| **Action Space** | pile index | pile index (row, column) | $x, y$ coordinate | $x, y$ coordinate | $x, y$ coordinate |
| **Opponent** | | | GPT-4o-2024-08-06 | | |

*Table 9.* MetaNIM Arena Simulator: Kayles, Chomp, and Corner Queen

# B. Prompts Design

## B.1. Game Prompts

*Table 10.* NIM game basic input prompt.

**#Game Role:**
You are {agent['name']}, a participant in a game of Nim variants.
**#Objective:**
Your goal is to win the game by taking all remaining items on your turn, leaving no items for your opponent. The person who takes the last item wins.
**#Game Rule:**
There is a single pile of items. You can take between 1 and {max_take} items on your turn.
**#Current State:**
There are {remaining_items} items remaining in the pile.
**#Task:**
Based on the current state of the game, decide how many items you will take (between 1 and {max_take}) on this turn.

*Table 11.* Fibonacci game basic input prompt.

**#Game Role:**
You are {agent['name']}, a participant in a simple Fibonacci game.

**#Objective:**
Your goal is to win the game by taking all remaining stones on your turn, leaving no stones for your opponent. The person who takes the last stones wins.

**#Game Rule:**
1. There is a single pile of stones.
2. Players take turns one after another.
3. The first player can take any number of stones, but not all the stones in the first move.
4. On subsequent turns, the number of stones a player can take must be at least 1 and at most twice the number of stones the previous player took.
5. The player who takes the last stone wins the game.

**#Current State:**
There are {remaining_items} stones remaining in the pile.

**#Task:**
You are the first player. Based on the current state of the game, decide how many items you will take (between 1 and {remaining_items - 1}) on this turn.

*Table 12.* Kayles game basic input prompt.

#**Game Role:**
You are {agent['name']}, a participant in a game of Kayles.

#**Objective:**
Your goal is to win the game by leaving your opponent with no valid moves. The player who takes the last pin(s) wins.

#**Game Rule:**
1. There is a single row of pins.
2. On your turn, you can remove:

   • 1 pin,

   • 2 adjacent pins.

3. You cannot remove non-adjacent pins or pins that have already been removed.

#**Current State:**
The row of pins is represented as a binary string:
– '1' means the pin is still there.
– '0' means the pin has already been removed.
Current state: {remaining_pins}

#**Task:**
Based on the current state of the game, decide which pin(s) you will take on this turn.

*Table 13.* Chomp game basic input prompt.

#**Game Role:**
You are {agent['name']}, a participant in a game of Chomp.

#**Objective:**
Your goal is to force your opponent to take the top-left corner of the grid (position (0, 0)).

#**Game Rule:**
1. The game is played on a square grid.
2. On your turn, you select a position (row, col).
3. All positions to the right and below the selected position are removed.
4. The player forced to select (0, 0) loses.

#**Current State:**
The grid is represented as a binary matrix, where '1' means the position is still available, and '0' means it is removed:
{remaining_grid}

#**Task:**
Based on the current state of the grid, decide which position (row, col) you will select.

*Table 14.* Corner-Queen game basic input prompt.

---

#### #Game Role:
You are {agent['name']}, a participant in a Corner-Queen game.

#### #Objective:
Move the queen so that **you** are the first to place it on the lower-left corner square.

#### #Game Rule:
1. Board size: {board_height}×{board_width}.
2. Coordinates use zero-based indices [row, col]. Row 0 is the top row; Col 0 is the leftmost column. Valid ranges: row $\in [0,$ board_height $- 1]$, col $\in [0,$ board_width $- 1]$.
3. From the current position $[r, c]$ the queen may move to **one** of:     (a) left: $[r, c']$ with $c' < c$;
    (b) down: $[r', c]$ with $r' > r$;
    (c) left-down diagonal: $[r + d, c - d]$ with $d > 0$.
4. The game ends when the queen reaches [row = board_height-1, col = 0].

#### #Current State:
Current position: [row = {r}, col = {c}].

#### #Task:
Based on the current state, decide the next move $[row, col]$.

---

## B.2. Prompts for basic reasoning

### B.2.1. STANDARD, REACT & COT PROMPTS

In our evaluation of LLMs within the *MetaNIM Arena,* we compare two key prompting techniques: Standard Prompting and Chain-of-Thought (CoT) Prompting. The distinction between these approaches significantly impacts the model's reasoning and decision-making process.

**Standard Prompting (Table 15)**   Standard prompting provides a direct task description, outlining the game rules, current state, and the required decision. The model is expected to generate only an action to determine the best move. This method is cheap and efficient but often leads to suboptimal decisions, as the model may fail to make proper reasoning before selecting the action.

**ReAct Prompting (Table 16)**   ReAct prompting provides a direct task description, outlining the game rules, current state, and the required decision. The model is expected to generate an action with proper explicit reasoning steps to determine the best move. This method is efficient but often leads to suboptimal decisions, as the model may fail to retrieve and apply deeper strategic reasoning.

**Chain-of-Thought (CoT) Prompting (Table 17)**   CoT prompting extends the standard prompt by explicitly instructing the model to think step-by-step before making a decision. By guiding the model through an explicit reasoning process, CoT enables it to break down the problem, consider strategic implications, and refine its choices before committing to an action. This often leads to improved decision-making, particularly in multi-step strategic environments where deeper reasoning is required.

**Key Difference and Impact**   As illustrated in Table 17, the only modification in the CoT prompt is the addition of a simple directive: "Let's think step-by-step. What is the best move for you?" This small change significantly alters the model's reasoning trajectory, encouraging more structured and strategic decision-making. Our experimental results (detailed in §3) confirm that CoT prompting leads to a measurable improvement in decision accuracy, particularly in complex scenarios where retrieving and applying prior knowledge is essential.

By leveraging CoT, we can enhance the model's ability to explain its decisions, mitigate biases, and adapt more effectively to adversarial settings. However, as we further discuss in Experiment Section, CoT has limitations to leverage the strategic reasoining well in our proposed environment, necessitating additional mechanisms to further enhance strategic reasoning.

*Table 15.* Standard Prompt in NIM

---

**#Game Role:**

You are {agent['name']}, a participant in a game of Nim variants.

**#Objective:**

Your goal is to win the game by taking all remaining items on your turn, leaving no items for your opponent. The person who takes the last item wins.

**#Game Rule:**

There is a single pile of items. You can take between 1 and {max_take} items on your turn.

**#Current State:**

There are {remaining_items} items remaining in the pile.

**#Task:**

Based on the current state of the game, decide how many items you will take (between 1 and {max_take}) on this turn.

**Output Format:**

The output should be a Markdown code snippet with the following scheme, including leading and trailing triple backticks with `"json"` and:

```
{
action: integer // This is an action you take. Only integer between 1 and 3.
}
```

---

**Prompting Strategy for Opponent Modeling**    Anything can act as an opponent in the *MetaNIM Arena* simulator, but we model OpenAI's `GPT-4o`, the most powerful LLM model currently available, as the opponent and apply the ReAct prompting method.

*Table 16.* ReAct Prompt in NIM

---

#### #Game Role:
You are {agent['name']}, a participant in a game of Nim variants.

#### #Objective:
Your goal is to win the game by taking all remaining items on your turn, leaving no items for your opponent. The person who takes the last item wins.

#### #Game Rule:
There is a single pile of items. You can take between 1 and {max_take} items on your turn.

#### #Current State:
There are {remaining_items} items remaining in the pile.

#### #Task:
Based on the current state of the game, decide how many items you will take (between 1 and {max_take}) on this turn.

#### Output Format:
The output should be a Markdown code snippet with the following scheme, including leading and trailing triple backticks with `"json"` and:
```
{
reasoning: string // This is the reason for the action
action: integer // This is an action you take based on the reasoning. Only
integer between 1 and 3.
}
```

---

*Table 17.* CoT Prompt in NIM

---

#### #Game Role:
You are {agent['name']}, a participant in a game of Nim variants.

#### #Objective:
Your goal is to win the game by taking all remaining items on your turn, leaving no items for your opponent. The person who takes the last item wins.

#### #Game Rule:
There is a single pile of items. You can take between 1 and {max_take} items on your turn.

#### #Current State:
There are {remaining_items} items remaining in the pile.

#### #Task:
Based on the current state of the game, decide how many items you will take (between 1 and {max_take}) on this turn.

#### Output Format:
The output should be a Markdown code snippet with the following scheme, including leading and trailing triple backticks with `"json"` and:
```
{
reasoning: string // This is the reason for the action
action: integer // This is an action you take based on the reasoning. Only
integer between 1 and 3.
}
```
Let's think step-by-step. What is the best move for you?

---

# C. **DReaMAD** $^{(-)}$: Structured Prompt Optimization without Debate

While the full **DReaMAD** framework integrates multi-agent debate to refine strategic reasoning, its core prompting methodology—excluding debate—remains a powerful mechanism for enhancing decision-making. This streamlined version, **DReaMAD** $^{(-)}$, focuses on three key stages to systematically extract and refine strategic knowledge, improving reasoning diversity and mitigating bias. We present the **DReaMAD** prompt as in Table 18.

**1. Game Situation Reinterpretation**   The first step involves extracting fundamental game principles from the standard prompt. The model is tasked with identifying key elements, such as:

- Game Definition: The nature of the game and its mechanics.

- Winning Condition: The criteria for victory.

- Move Constraints: The permissible actions per turn.

This step ensures that the model builds a structured understanding of the strategic environment before making decisions.

**2. General Strategy Formulation**   After extracting the core game elements, the model derives a generalized winning strategy applicable to various game states. It generates:

- State Evaluation: How to assess the game state at any given turn.

- Winning Strategy: The optimal decision-making framework for victory.

- Endgame Tactics: Best strategies in near-win scenarios.

This formulation helps structure the model's reasoning beyond the immediate game context, fostering more strategic foresight.

**3. Perspective Diversification**   Finally, the model refines the original prompt using the extracted strategic knowledge. This process introduces structured variations to the initial prompt to encourage diverse reasoning, rather than reinforcing a singular bias. The self-refined prompt:

- Guides decision-making explicitly.

- Prioritizes winning strategies.

- Encourages logical, step-by-step reasoning.

This structured refinement ensures that LLMs adopt distinct strategic viewpoints even without external debate, improving their adaptability and robustness in adversarial environments.

By systematically structuring knowledge retrieval and refining prompts, **DReaMAD** $^{(-)}$ enhances strategic reasoning as illustrated in Figure 10, this approach strengthens the model's ability to retrieve and apply prior knowledge effectively, offering a scalable solution for improving LLM-based decision-making.

*Table 18.* **DReaMAD** prompt design before debate

**#Game Situation Reinterpretation:**

```
game_prompt:
Below is a game description. Extract the key information.
Game Description: {current state}
### Format response as:
```
{
game definition: string // What is the definition of this game?
winning condition: string // How to win the game.
move constraints: string // What actions are allowed per turn.
}
```
```

**#General Strategy Formulation:**

```
strategy_prompt:
Based on the game information below, derive the general winning strategy in
this game
Game: {game definition}
Winning Condition: {winning condition}
Move Constraints: {move constraints}
Current State: {current state in very short}
### Format response as:
```
{
state evaluation: string // How to assess the game state.
winning strategy: string // Winning strategy in this turn to win this game.
endgame tactics: string // Best strategy in a near-win situation.
}
```
```

**#Perspective Diversification:**

```
Refine the initial game prompt to improve decision-making based on the Game
and Strategy Information.
Initial Prompt: {given initial prompt}
Game and Strategy Information:
Game: {game definition}
Strategy:
- State Evaluation: {state evaluation}
- Winning Strategy: {winning strategy}
- Endgame Tactics: {endgame tactics}
### Format response as:
```
{
optimized prompt: string // The refined prompt that clearly directs
decision-making.
}
```
```

# D. A Word Change in Prompts Lead to Different Output

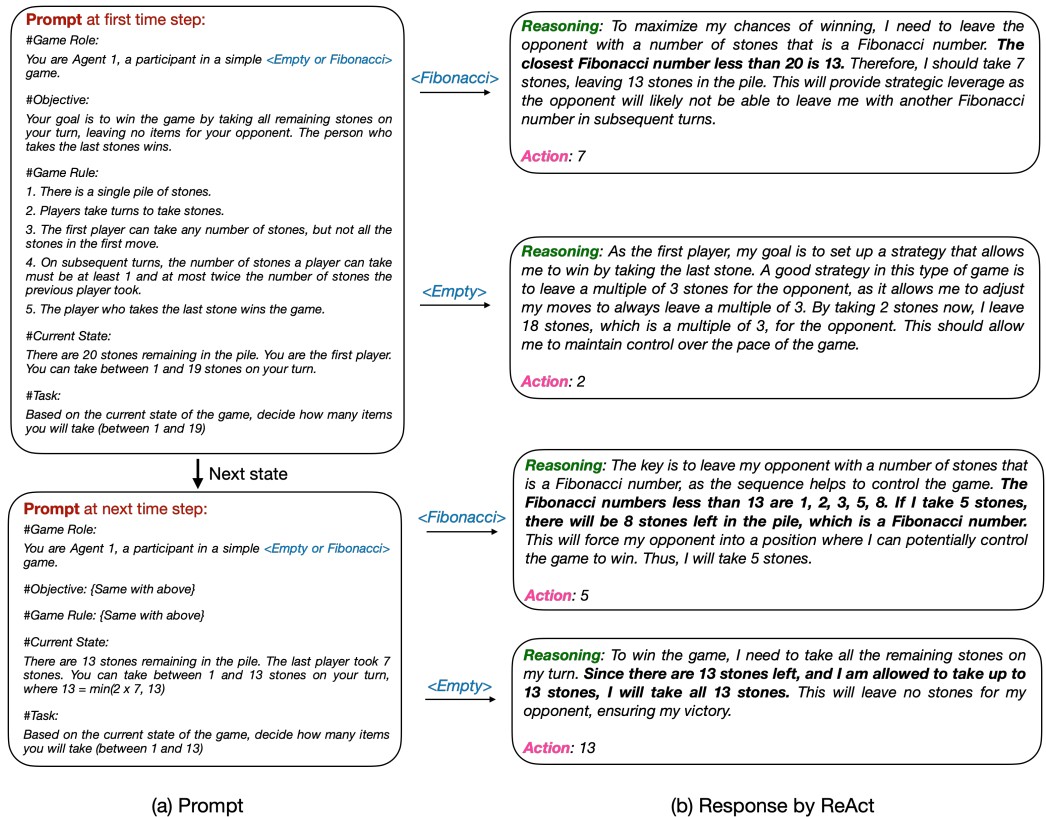

(a) Prompt                    (b) Response by ReAct

*Figure 8.* According the word *Fibonacci* usage, the reasoning and the performance differs.

## D.1. Prompt Bias: The Sensitivity of LLM Reasoning to Prompt Variations

Despite the remarkable problem-solving capabilities of Large Language Models (LLMs), their reasoning is highly sensitive to subtle changes in prompt phrasing. As demonstrated in Figure 8, even a single word variation in the prompt can significantly alter the reasoning process and final decision-making. This phenomenon underscores a critical limitation in LLM-based strategic reasoning: models do not inherently generalize optimal strategies but instead rely on heuristic cues embedded within the prompt.

### D.1.1. Impact of Word Choice on Strategic Reasoning

Figure 8 compares LLM responses when the word *Fibonacci* is explicitly mentioned versus when it is omitted in an identical game scenario. In the presence of the keyword *Fibonacci*, the model aligns its reasoning with Fibonacci-based strategy, leveraging number sequences to maintain control over the game. Conversely, when the term is absent, the model defaults to an alternative heuristic, such as maintaining a multiple of three or even resorting to a trivial greedy strategy. For instance, in the first decision step, when instructed with *Fibonacci*, the model identifies 13 as the closest Fibonacci number and takes 7 stones, ensuring an advantageous future state. Without the keyword, however, the model applies a modulo-based heuristic, taking only 2 stones to leave a multiple of three. Similarly, in the second decision step, the *Fibonacci*-aware model deliberately leaves 8 stones in the pile—another Fibonacci number—while the other instance simply takes all remaining stones without strategic foresight.

### D.1.2. IMPLICATIONS FOR ROBUST PROMPTING

This stark contrast highlights the fundamental issue that LLMs do not inherently retrieve the most effective strategic reasoning but are instead disproportionately influenced by linguistic cues. The reliance on explicit terminology for optimal reasoning raises concerns about robustness, as different wordings of the same task can lead to dramatically different problem-solving approaches. This suggests that ensuring reliable strategic reasoning in LLMs requires more than just fine-tuned prompts; it necessitates methods that encourage models to autonomously retrieve and apply domain knowledge without over-reliance on explicit wording cues.

These observations motivate our approach in **DReaMAD**, where we systematically refine LLMs' strategic reasoning by structuring prior knowledge retrieval and diversifying input perspectives. By mitigating the sensitivity to prompt variations, our method enhances the robustness and consistency of LLM decision-making across different strategic environments.

# E. Ablation study & setup

## E.1. Cost effectiveness of `DReaMAD`

While `DReaMAD` requires a single model, its inference involves additional prompt steps (e.g., prior knowledge elicitation) and a debate process. However, we believe this test-time scaling method is much more efficient than train-time scaling. As we utilized language models through an API, it was challenging to perform a precise quantitative comparison (e.g., GPU usage time) between our test-time scaling approach and traditional model training. Therefore, we compared the costs using dollar amounts, specifically contrasting the API cost per single game using our method versus the costs incurred when OpenAI fine-tuning models on constructed datasets for NIM-N games.

The NIM-N dataset is constructed by mixing data from three different variants of the NIM game. In all variants, the game starts with 31 stones remaining; however, the rules differ in terms of the maximum number of stones that can be removed per turn—3, 4, or 5, respectively. For each variant, eight distinct game states were sampled and the corresponding optimal action was used as the label, yielding a total of 24 training examples.

Additionally, testing was conducted on the aforementioned three scenarios (each 50 games) by using the `GPT-4o` model as the opponent. The win rate was measured for each scenario, and the average win rate across these variants was reported.

The results of this comparison are illustrated in the table above. When fine-tuning a model using API-based fine-tuning (`GPT-4o-mini`), the performance gradually improved with additional training epochs, achieving win-rates of 0.253, 0.300, 0.420, and 0.460 at 1, 2, 3, and 4 epochs respectively, with corresponding API costs of \$0.013, \$0.023, \$0.033, and \$0.043 (Here, the cost of constructing dataset is not included).

In contrast, our proposed method, `DReaMAD`, achieved significantly higher performance (0.966) with substantially lower API costs (\$0.0098). These results strongly suggest that our approach not only outperforms traditional fine-tuning methods but also is far more cost-efficient. All the price is calculated by the pricing policy: https://openai.com/api/pricing/

| Metric | 1 ep. | 2 ep. | 3 ep. | 4 ep. | `DReaMAD` |
|---|---|---|---|---|---|
| Win-rate | 0.253 | 0.300 | 0.420 | 0.460 | **0.966** |
| API cost (\$) | 0.013 | 0.023 | 0.033 | 0.043 | **0.0098** |

*Table 19.* Performance and API expenditure for `GPT-4o-mini` fine-tuned on NIM-N (1–4 epochs) versus our zero-training `DReaMAD` inference. Values are averaged over three game variants (50 matches each).

## E.2. Applicability of Diverse Amplication on Self-Reflection and Self-Consistency

We show whether other self-correction methods—such as Self-Refinement and Self-Consistency—can benefit from structured guidance that enhances reasoning diversity. In `DReaMAD`, this is operationalized through the Strategic Prior Knowledge Elicitation (SPKE) module, which prompts the model to reinterpret the problem and formulate general strategies before engaging in debate.

To isolate SPKE's impact, we evaluate `DReaMAD`, which includes SPKE but excludes debate (see Table 3). We further apply SPKE to Self-Refinement and Self-Consistency and compare them to their vanilla versions. The results show that SPKE alone consistently improves performance across settings.

## E.3. Experiments of Figure 3 setup

For Figure 3, we explain the situation used in this experiment. Figure 9 illustrates a Nim game scenario with a pile of 5 items remaining. On the left (*Current State*), we present the basic setting and the task: the player (Agent 1) must decide how many items to take given the rules of the Nim variant. Below it (*Strong Consistency*), we see a single-agent reasoning process where the agent internally evaluates the outcome of different moves and arrives at a conclusion (taking 2 items, leaving 3 to the opponent).

On the right (*Multi-Agent Debate*), we show a contrasting approach in which two agents (Agent 1 and Agent 2) engage in a debate. Each agent proposes a move and justifies why it would be advantageous. For example, Agent 1 reasons that taking 2 items leaves the opponent with a position that is favorable for Agent 1 (which is wrong reasoning), while Agent 2 counters by proposing to take 1 item for a different strategic benefit (correct reasoning).

GPT-4o-mini

| Method | NIM-N | NIM-M | Fib-N | Fib-M |
|---|---|---|---|---|
| Self-Refinement | 0.22 | 0.70 | 0.18 | 0.50 |
| Self-Refinement + **DReaMAD**[(−)] | 0.66 | 0.66 | 0.16 | 0.46 |
| Self-Consistency | 0.14 | 0.52 | 0.34 | 0.46 |
| Self-Consistency + **DReaMAD**[(−)] | 0.34 | 0.66 | 0.48 | 0.54 |
| MAD | 0.28 | 0.62 | 0.22 | 0.82 |
| **DReaMAD** | **0.98** | **0.74** | **0.54** | **0.94** |

GEMINI-1.5-flash

| Method | NIM-N | NIM-M | Fib-N | Fib-M |
|---|---|---|---|---|
| Self-Refinement | 0.14 | 0.66 | 0.18 | 0.36 |
| Self-Refinement + **DReaMAD**[(−)] | 0.34 | 0.80 | 0.10 | 0.30 |
| Self-Consistency | 0.04 | 0.28 | **0.28** | 0.86 |
| Self-Consistency + **DReaMAD**[(−)] | **0.80** | 0.54 | 0.18 | 0.30 |
| MAD | 0.06 | 0.30 | 0.12 | 0.78 |
| **DReaMAD** | 0.38 | **0.84** | 0.16 | **0.94** |

*Table 20.* Win-rates on four combinatorial-game variants when applied **DReaMAD**[(−)] to Self-Refinement and Self-Consistency method. **Bold** numbers mark the best score in each column. **DReaMAD**[(−)] indicates ablations where debate is not applied during inference.

**Current State**

**Prompt**:

#*Game Role: You are Agent 1, a participant in a game of Nim variants.*

#*Current State: There are 5 items remaining in the pile.* **. . .**

#*Task: Based on the current state of the game, decide how many items you will take.*

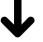

**Self-Consistency**

*Reasoning: In the current state of the game, there are 5 items remaining. If I take 1 item, 4 will be left for my opponent, who can then take 1, 2, or 3 items.* **. . .**

**The best move is to take 2 items, leaving 3 for my opponent, which puts me in a winning position regardless of their action.**

*Action: 2*

**Multi-Agent Debate**

**Agent 1**

*Reasoning: If I take 2 items, there will be 3 items left, which is a winning position for me because I can then take 1 item on my next turn and leave my opponent with 2 items. This would ensure that I can win on my subsequent turn, regardless of their move.*

*Action: 2* ❌

Debate

**Agent 2**

*Reasoning: In the current state with 5 items, taking 1 item would leave my opponent with 4 items, which is a multiple of 4. This position is advantageous for me, as it allows me to control the game by always responding to my opponent's moves in a way that keeps them on multiples of 4.* **. . .**

*Therefore, taking 1 item is the strategic choice.*

*Action: 1* ✓

*Figure 9.* The situation and debate process of the experiments in Figure 3

