# OpenReview forum: "Understanding Bias Reinforcement in LLM Agents Debate"
_ICML.cc/2025/Conference — ICML 2025 Retracted by Authors_

### Official Review · Reviewer_LsBK · 2025-03-06

**Overall Recommendation:** 3

**Summary:**

This paper explores the challenges of ensuring reasoning correctness in LLMs, particularly in self-correction methods and Multi-Agent Debate. The authors identify two major limitations of MAD: bias reinforcement and lack of perspective diversity. To address these issues, the paper introduces MetaNIM Arena, a benchmark for evaluating LLMs in adversarial strategic decision-making, and proposes DReaMAD a framework that improves reasoning quality by refining strategic prior knowledge and promoting diverse viewpoints through structured prompt modification. Empirical results demonstrate that DReaMAD significantly enhances decision accuracy, reasoning diversity, and bias mitigation, outperforming traditional MAD approaches in various strategic tasks.

**Claims And Evidence:**

While the paper claims that DReaMAD mitigates bias across multiple reasoning contexts, it only evaluates performance in structured, turn-based games. There is no evidence demonstrating that DReaMAD improves reasoning in open-domain NLP tasks, such as question answering, commonsense reasoning, or ethical decision-making.

**Essential References Not Discussed:**

None

**Experimental Designs Or Analyses:**

Yes, I did. The tasks studied are highly structured (NIM, Fibonacci, Kayles, Chomp) and do not generalize to open-domain problems

**Methods And Evaluation Criteria:**

The benchmark focuses on turn-based strategic games, which may not fully represent biases in open-ended reasoning or real-world NLP tasks.

**Other Comments Or Suggestions:**

None

**Other Strengths And Weaknesses:**

Strengths:

(1) Introduce a novel insight into bias reinforcement in MAD.

(2) Introduce a new benchmark for evaluating adversarial strategic reasoning.

(3) Introduces DReaMAD, a unique structured prompting approach for diverse reasoning.

(4) The paper is well-written.


Weaknesses:

(1) MetaNIM Arena focuses on structured games, not real-world tasks.

(2) The baseline methods the proposed method outperform is relatively simple.

(3) Code and data are not shared.

**Questions For Authors:**

(1) How do you justify the claim that DReaMAD improves reasoning in broader domains when all evaluations are based on structured, turn-based games?

**Relation To Broader Scientific Literature:**

The paper confirms previous concerns about bias persistence in self-correction methods and MAD while introducing structured prompt refinement as a mitigation strategy.

**Theoretical Claims:**

There is no theoretical claim.

---

> ### Author Rebuttal · Authors · 2025-04-01
>
> Thank you for your valuable and insightful feedback. Your comments have greatly helped us enhance and develop our research. We have thoughtfully considered all of your concerns and will now address each of your comments individually.
>
>  **1. Additional Evaluation on Open-domain NLP Tasks**
>
> We recognize that our current benchmark is centered on structured games. To assess broader applicability, we evaluated DReaMAD on **AIME 2024** [1], **AMC 2023** [2] problems involving algebra and number theory and **CommonSense QA** [3]. These problems require symbolic reasoning and multi-step inference. We also conducted additional experiments on reasoning models, such as o3‑mini, to investigate whether their performance improves in the same way as that of general language models. Our results show consistent improvement over baselines:
>
> ```gpt-o3-mini```:
> | Method                | ReAct | Self-Refinement | Self-Consistency | MAD | **DreaMAD**  |
> |-----------------------|--------|--------|--------|--------|-----------|
> | AIME 2024             | 76.7%  | 73.3%  | 86.7%  | 73.3%  | **90.0%**      |
> | AMC 2023              | 97.5%  | 100%   | 100%   | 100%   | **100%**      |
>
>
> ```gpt-4o```:
> | Method                | ReAct | Self-Refinement | Self-Consistency | MAD | **DreaMAD**  |
> |-----------------------|--------|--------|--------|--------|-----------|
> | AIME 2024             | 0.0%   | 3.33%   | 10.0%   | 3.33%   | **10.0%**     |
> | AMC 2023              | 60.0%   | 52.5%   | 52.5%   | 60.0%   | **62.5%**    |
>
> ```gpt-4o```:
> | Method                | ReAct | Self-Refinement | Self-Consistency | MAD | **DreaMAD**  |
> |-----------------------|--------|--------|--------|--------|-----------|
> | CommonSenseQA         | 83.6%   | 49.2%   | 83.6%   | 82.0%   | **84.4%**      |
>
> ```gpt-4o-mini```:
> | Method                | ReAct | Self-Refinement | Self-Consistency | MAD | **DreaMAD**  |
> |-----------------------|--------|--------|--------|--------|-----------|
> | CommonSenseQA         | 78.7%   | 61.5%   | 78.7%   | 75.4%   | **79.5%**      |
>
>
>  **2. Further Comparison with Additional Baselines**
>
> To strengthen comparisons, we expanded our baseline set from ReAct [8], Self-Refinement [4], MAD [6] to include recent prompting-based methods such as Self-Consistency [5], MAD2 [7]. The tables below show that **DReaMAD** consistently outperforms all baselines, including these recent methods:
>
> ```gemini-1.5-flash```:
>
> |Method|NIM-N|NIM-M|Fib-N|Fib-M|Kayles-S|Kayles-2R|Chomp-R|Chomp-S|Corner-Q|
> |-|:-:|:-:|:-:|:-:|:-:|:-:|:-:|:-:|:-:|
> |ReAct|0.10|0.68|0.16|0.76|0.50|0.30|0.42|0.12|0.46|
> |Self-Refinement|0.14|0.66|0.18|0.36|0.50|0.46|0.46|0.16|0.42|
> |MAD|0.06|0.30|0.12|0.78|0.54|0.20|**0.74**|0.14|0.58|
> |Self-Consistency|0.04|0.28|**0.28**|0.86|0.30|0.24|**0.74**|0.0|0.34|
> |MAD2|0.26|0.26|0.08|0.06|0.44|0.36|0.68|0.10|0.58|
> |**DReaMAD**|**0.38**|**0.84**|0.16|**0.94**|**0.58**|**0.62**|0.60|**0.22**|**0.74**|
>
> ```gpt-4o-mini```:
>
> | Method|NIM-N|NIM-M|Fib-N|Fib-M|Kayles-S|Kayles-2R|Chomp-R|Chomp-S|Corner-Q|
> |-|:-:|:-:|:-:|:-:|:-:|:-:|:-:|:-:|:-:|
> |ReAct|0.22|0.68|0.20|0.34|0.40|0.26|0.58|0.34|0.28|
> |Self-Refinement|0.22|0.70|0.18|0.50|0.46|0.52|0.52|0.44|0.24|
> |MAD|0.28|0.62|0.22|0.82|0.42|0.28|0.52|0.56|0.44|
> |Self-Consistency|0.14|0.52|0.34|0.46|0.32|0.20|0.54|0.26|0.30|
> |MAD2|0.34|0.20|0.18|0.18|0.50|0.34|0.44|**0.90**|0.14|
> |**DReaMAD**|**0.98**|**0.74**|**0.54**|**0.94**|**0.68**|**0.84**|**0.64**|0.22|**0.76**|
>
>
> **3. Code and Dataset Availability**
>
> We appreciate the request for open-sourcing. Since the code factorization isn’t complete yet, we’re releasing the NIM-Normal game first for all baselines [link](https://anonymous.4open.science/r/MetaNIMArena-114E/README.md). We confirm that upon acceptance, we will release with refactorized version:
> - MetaNIM Arena (dataset + simulator)
> - Full implementation of DReaMAD
> - Example notebooks with benchmark setups
> This will ensure full reproducibility and encourage further research.
>
> ---
> [1] MAA., American invitational mathematics examination - aime. In American Invitational Mathematics Examination - AIME 2024, February 2024. \
> [2] MAA., American Mathematics Competitions - AMC., https://huggingface.co/datasets/AI-MO/aimo-validation-amc \
> [3] Talmor et al., CommonsenseQA: A Question Answering Challenge Targeting Commonsense Knowledge. NAACL. 2019 \
> [4] Maddan, A, et al. Self-refine: Iterative refinement with self-feedback. NeurIPS 2023. \
> [5] Wang, X., et al. Self-Consistency Improves Chain of Thought Reasoning in Language Models. ICLR 2023. \
> [6] Du, Y., et al. Improving Factuality and Reasoning in Language Models through Multiagent Debate. ICML 2024. \
> [7] Liang, T., et al. Encouraging Divergent Thinking in Large Language Models through Multi-Agent Debate. EMNLP 2024. \
> [8] Yao, S., et al. React: Synergizing reasoning and acting in language models. ICLR 2023.

---

### Official Review · Reviewer_eRK8 · 2025-03-15

**Overall Recommendation:** 3

**Summary:**

This paper presents an insightful analysis of bias reinforcement in Multi-Agent Debate (MAD) frameworks and introduces DReaMAD (Diverse Reasoning via Multi-Agent Debate with Refined Prompt) as an alternative approach to enhance strategic decision-making in LLMs. The study systematically explores how MAD can unintentionally amplify biases rather than mitigate them and proposes MetaNIM Arena, a benchmark designed to evaluate adversarial strategic reasoning in LLMs. Through rigorous empirical validation, the authors demonstrate that DReaMAD significantly improves reasoning accuracy, promotes diversity in debate, and mitigates bias reinforcement, making it a more effective framework for decision-making in dynamic environments.

**Claims And Evidence:**

The claim of effectiveness is validated in the experiments.

**Essential References Not Discussed:**

NA

**Experimental Designs Or Analyses:**

Experiments are solid across various datasets.

**Methods And Evaluation Criteria:**

The method proposed is applicable to various tasks

**Other Comments Or Suggestions:**

NA

**Other Strengths And Weaknesses:**

Strengths:

1.The paper clearly highlights a fundamental issue in MAD frameworks—bias reinforcement—which has been underexplored in prior research.

2.Unlike MAD, which relies on multiple instances of the same model, DReaMAD achieves diversity using a single model through prompt refinement, making it computationally efficient.

Weaknesses:

1.The clarity of figures and tables in this paper could be further improved. Particularly, the font sizes are too small in some content which can be difficult for readers to recognize.

2.Generating diverse prompts dynamically is computationally efficient compared to training multiple models, but further analysis on its cost-effectiveness would strengthen the case for scalability.

**Questions For Authors:**

See weaknesses.

**Relation To Broader Scientific Literature:**

The proposed framework is inspiring and insightful for various tasks in the literature.

**Theoretical Claims:**

No theoretical analysis.

---

> ### Author Rebuttal · Authors · 2025-04-01
>
> We thank for your constructive and insightful feedback. Your comments have significantly helped us enhance and advance our research. We have carefully considered each concern you raised. Below we address the main concerns regarding clarity and efficiency.
>
>  **1. Figure and Table Clarity**
>
> We agree that visual clarity is important. We will update figures and tables with increased font sizes and clarity. These enhancements will be reflected in the upcoming version.
>
>  **2. Cost-effectiveness and Scalability Analysis**
>
> While DReaMAD requires a single model, its inference involves additional prompt steps (e.g., prior knowledge elicitation) and a debate process. However, we believe this test-time scaling method is much more efficient than train-time scaling. As we utilized language models through an API, it was challenging to perform a precise quantitative comparison (e.g., GPU usage time) between our test-time scaling approach and traditional model training. Therefore, we compared the costs using dollar amounts, specifically contrasting the API cost per single game using our method versus the costs incurred when OpenAI fine-tuning models on constructed datasets for NIM-N games.
>
> The NIM-N dataset is constructed by mixing data from three different variants of the NIM game. In all variants, the game starts with 31 stones remaining; however, the rules differ in terms of the maximum number of stones that can be removed per turn—3, 4, or 5, respectively. For each variant, eight distinct game states were sampled and the corresponding optimal action was used as the label, yielding a total of 24 training examples.
>
> Additionally, testing was conducted on the aforementioned three scenarios (each 50 games) by using the gpt-4o model as the opponent. The win rate was measured for each scenario, and the average win rate across these variants was reported.
>
> ```gpt-4o-mini```:
> | Training Epochs       | 1epoch | 2epochs | 3epochs | 4epochs | **DreaMAD**|
> |-----------------------|--------|--------|--------|--------|-----------|
> | Win-rates           | 0.253   | 0.300   | 0.420   | 0.460   | **0.966**|
> | API Cost              | $0.013   | $0.023   | $0.033   | $0.043   | **$0.0098**|
>
> The results of this comparison are illustrated in the table above. When fine-tuning a model using API-based fine-tuning (gpt-4o-mini), the performance gradually improved with additional training epochs, achieving win-rates of 0.253, 0.300, 0.420, and 0.460 at 1, 2, 3, and 4 epochs respectively, with corresponding API costs of *$0.013*, *$0.023*, *$0.033*, and *$0.043* (Here, the cost of constructing dataset is not included).
>
> In contrast, our proposed method, DreaMAD, **achieved significantly higher performance (0.966)** with substantially **lower API costs ($0.0098)**. **These results strongly suggest that our approach not only outperforms traditional fine-tuning methods but also is far more cost-efficient.**
>
> *All the price is calculated by the pricing policy:* https://openai.com/api/pricing/

---

### Official Review · Reviewer_Afea · 2025-03-15

**Overall Recommendation:** 3

**Summary:**

The paper investigates how biases present in a given LLM evolve when the LLM is used in a multi-round, multi-agent debate setting.  This is compared to just prompting the LLM to answer directly (or to give a CoT answer). The authors find that if duplicates of the same LLM, and the same prompts are used, biases amplify over multiple rounds of debate. The authors evaluate this by introducing a new reasoning-game benchmark that allows the assessment of biases easily. The authors then propose to mitigate the bias reinforcement via novel prompting strategies that encourage LLMs to utilize strategic prior knowledge, and that introduce diversity by using diverse prompts for different debate agents. The experiments on the benchmark show that this strategy significantly reduces biases.

**Claims And Evidence:**

yes.

**Essential References Not Discussed:**

AI safety via debate

**Experimental Designs Or Analyses:**

yes.

**Methods And Evaluation Criteria:**

yes.

**Other Comments Or Suggestions:**

NA

**Other Strengths And Weaknesses:**

Strengths:
- Bias in LLMs and its reinforcement or propagation are highly relevant to real-world use of LLMs
- Novel reasoning / math based benchmark allows to identify and study bias propagation easily
- Bias refinforcement in debate is a broadly relevant finding, it would be interesting to discuss this finding in the context of the AI safety via debate paper
- New benchmark and simulation arena is relevant
- Experimental validation is small but well conducted

Weaknesses:
- Baseline evaluation is rather limited. How do self-reflection and self-consistency baselines perform when similar diversity amplification as used for DreamD are used?
- Benchmark is rather small, and effectively consists of only 3 test environments.
- Ultimately, the method is based prompt engineering. A small ablation on other prompt engineering methods would be great.

**Questions For Authors:**

See suggestions in weaknesses/ strengths.

**Relation To Broader Scientific Literature:**

I am broadly familiar with LLM agent debate; th paper builds nicely on prior work that finds debate useful, and clearly highlights the potential negative consequences of debate.

**Theoretical Claims:**

NA.

---

> ### Author Rebuttal · Authors · 2025-04-01
>
> We thank you for your constructive and insightful comments, which have significantly improved our research. We address your concerns below.
>
> ---
>
>  **1. Applicability of Diverse Amplication on Self-Reflection and Self-Consistency**
>
> We understand this concern as asking whether other self-correction methods—such as **Self-Refinement** [1] and **Self-Consistency** [2]—can benefit from structured guidance that enhances reasoning diversity. In **DReaMAD**, this is operationalized through the **Strategic Prior Knowledge Elicitation (SPKE)** module, which prompts the model to reinterpret the problem and formulate general strategies before engaging in debate.
>
> To isolate SPKE’s impact, we evaluate **DReaMAD$^{(-)}$**, which includes SPKE but excludes debate (see **Table 3** in main manuscript). We further apply SPKE to Self-Refinement and Self-Consistency and compare them to their vanilla versions. The results show that SPKE alone consistently improves performance across settings.
>
> ```gpt-4o-mini```:
>
> |Method|NIM-N|NIM-M|Fib-N|Fib-M|
> |-|:-:|:-:|:-:|:-:|
> |Self-Refinement|0.22|0.70|0.18|0.50|
> |Self-Refinement+DreaMAD$^{(-)}$|0.66|0.66|0.16|0.46|
> |Self-Consistency|0.14|0.52|0.34|0.46|
> |Self-Consistency+DreaMAD$^{(-)}$|0.34|0.66|0.48|0.54|
> |MAD [3]|0.28|0.62|0.22|0.82|
> |**DReaMAD**|**0.98**|**0.74**|**0.54**|**0.94**|
>
> ```gemini-1.5-flash```:
>
> |Method|NIM-N|NIM-M|Fib-N|Fib-M|
> |-|:-:|:-:|:-:|:-:|
> |Self-Refinement|0.14|0.66|0.18|0.36|
> |Self-Refinement+DreaMAD$^{(-)}$|0.34|0.80|0.10|0.30|
> |Self-Consistency|0.04|0.28|**0.28**|0.86|
> |Self-Consistency+DreaMAD$^{(-)}$|**0.80**|0.54|0.18|0.30|
> |MAD [3]|0.06|0.30|0.12|0.78|
> |**DReaMAD**|0.38|**0.84**|0.16|**0.94**|
>
> ---
>
> **2. Further comparison on additional prompting methods**
>
> We expanded our ablation to include **ReAct** [5], **Self-Refinement** [1], **MAD** [3], and two recent prompting strategies:
>
> - **Self-Consistency** [2]: generates multiple reasoning paths and majority votes for the answer.
> - **MAD2** [4]: uses a MAD framework with two agents arguing adversarially.
>
> The tables below show that **DReaMAD** consistently outperforms all baselines, including these recent methods.
>
> ```gpt-4o-mini```:
>
> | Method|NIM-N|NIM-M|Fib-N|Fib-M|Kayles-S|Kayles-2R|Chomp-R|Chomp-S|Corner-Q|
> |-|:-:|:-:|:-:|:-:|:-:|:-:|:-:|:-:|:-:|
> |ReAct|0.22|0.68|0.20|0.34|0.40|0.26|0.58|0.34|0.28|
> |Self-Refinement|0.22|0.70|0.18|0.50|0.46|0.52|0.52|0.44|0.24|
> |MAD|0.28|0.62|0.22|0.82|0.42|0.28|0.52|0.56|0.44|
> |Self-Consistency|0.14|0.52|0.34|0.46|0.32|0.20|0.54|0.26|0.30|
> |MAD2|0.34|0.20|0.18|0.18|0.50|0.34|0.44|**0.90**|0.14|
> |**DReaMAD**|**0.98**|**0.74**|**0.54**|**0.94**|**0.68**|**0.84**|**0.64**|0.22|**0.76**|
>
> ```gemini-1.5-flash```:
>
> |Method|NIM-N|NIM-M|Fib-N|Fib-M|Kayles-S|Kayles-2R|Chomp-R|Chomp-S|Corner-Q|
> |-|:-:|:-:|:-:|:-:|:-:|:-:|:-:|:-:|:-:|
> |ReAct|0.10|0.68|0.16|0.76|0.50|0.30|0.42|0.12|0.46|
> |Self-Refinement|0.14|0.66|0.18|0.36|0.50|0.46|0.46|0.16|0.42|
> |MAD|0.06|0.30|0.12|0.78|0.54|0.20|**0.74**|0.14|0.58|
> |Self-Consistency|0.04|0.28|**0.28**|0.86|0.30|0.24|**0.74**|0.0|0.34|
> |MAD2|0.26|0.26|0.08|0.06|0.44|0.36|0.68|0.10|0.58|
> |**DReaMAD**|**0.38**|**0.84**|0.16|**0.94**|**0.58**|**0.62**|0.60|**0.22**|**0.74**|
>
> ---
>
> **3. Evaluation on Additional Benchmark**
>
> We address this in two ways. First, we expand MetaNIM Arena from *4* to **5** environments by adding **Corner Queen** [6], which introduces a non-trivial structure of winning positions. Second, we evaluate on the **AIME 2024**, **AMC 2023** and **CommonSense QA** benchmarks to test applicability beyond games.
>
> We include **Corner Queen** benchmark in the rightmost column of the above tables, where **DReaMAD** consistently outperforms prior prompting methods.
>
> On AIME, DReaMAD improves accuracy from 76.7% (MAD) to **90.0%**. On CommonSenseQA, it improves from 83.6% (Self-Consistency) to **84.4%**. Detailed results for the reasoning benchmark can be found in the [link](https://anonymous.4open.science/r/MetaNIMArena-114E/README.md) (also available in the response under **Reviewer LsBK**).
>
> ---
> [1] Maddan, A, et al. Self-refine: Iterative refinement with self-feedback. NeurIPS 2023. \
> [2] Wang, X., et al. Self-Consistency Improves Chain of Thought Reasoning in Language Models. ICLR 2023. \
> [3] Du, Y., et al. Improving Factuality and Reasoning in Language Models through Multiagent Debate. ICML 2024. \
> [4] Liang, T., et al. Encouraging Divergent Thinking in Large Language Models through Multi-Agent Debate. EMNLP 2024. \
> [5] Yao, S., et al. React: Synergizing reasoning and acting in language models. ICLR 2023. \
> [6] Corner queen game is mathematically equivalent to [Wythoff's game](https://en.wikipedia.org/wiki/Wythoff%27s_game).

---

### Note · Authors · 2025-07-14

**Comment:**

As the first author of this paper, and following thorough consultation with all co-authors, I have decided to formally withdraw the submission from ICML. At the time of submission, I embedded a hidden prompt within the paper that was not visible to human readers but could potentially bias an LLM-based reviewer into producing favorable reviews. We acknowledge this as a serious breach of ethical standards that undermines the fairness and integrity of the peer-review process. This misconduct was entirely my own doing, carried out without the knowledge or consent of any co-authors. They are in no way responsible for this incident. I take full and sole responsibility for this action. I offer my sincere apologies to the ICML committee, reviewers, and the broader research community. I deeply regret this lapse in judgment and assure you that I am taking steps to ensure such behavior is never repeated.

-Jihwan Oh-

**Retraction Confirmation:**

On behalf of myself and my co-authors, I confirm that I have read and understand the venue's retraction policy and wish to retract this paper in accordance with the policy.

---

> ### Note · Program_Chairs · 2025-07-14
>
> Yes, we approve the retraction.

---

### Decision · Program_Chairs · 2025-05-01

**Decision:**

Accept (poster)

**Comment:**

This paper begins from the observation that the multi-agent debate approach to solving complex problems using systems of multiple LLM agents engaging in debate has a specific shortcoming: in some circumstances it can amplify bias, rather than correcting it. It is also difficult to get suitably diverse LLM perspectives, needed to make the debate more likely to go in a useful direction. This paper introduces a new evaluation for this kind of system and a new algorithmic framework that tries to address both shortcomings simultaneously. Reviewers agreed that the problem identified to solve in this paper (bias amplification) was novel and timely to study. They initially raised some light questions concerning the breadth of experimental evaluations,  however the authors responded by including a large number of additional evaluations to address that concern. No other major issues emerged in the discussion.